# Augmentation of progestin signaling rescues testis organization and spermatogenesis in zebrafish with the depletion of androgen signaling

Gang Zhai[1,2†], Tingting Shu[1,2,3†], Guangqing Yu[1,2], Haipei Tang[4], Chuang Shi[1,2], Jingyi Jia[5], Qiyong Lou[1], Xiangyan Dai[6], Xia Jin[1], Jiangyan He[1], Wuhan Xiao[1,2,7], Xiaochun Liu[4], Zhan Yin[1,2,7]*

[1]State Key Laboratory of Freshwater Ecology and Biotechnology, Chinese Academy of Sciences, Wuhan, China; [2]College of Advanced Agricultural Sciences, University of Chinese Academy of Sciences, Beijing, China; [3]Chinese Sturgeon Research Institute, China Three Gorges Corporation, Hubei, China; [4]5State Key Laboratory of Biocontrol, Institute of Aquatic Economic Animals and Guangdong Provincial Key Laboratory of Improved Variety Reproduction in Aquatic Economic Animals, School of Life Sciences, Sun Yat-Sen University, Guangzhou, China; [5]College of Fisheries, Huazhong Agriculture University, Wuhan, China; [6]Key Laboratory of Freshwater Fish Reproduction and Development and Key Laboratory of Aquatic Science of Chongqing, School of Life Science, Southwest University, Chongqing, China; [7]The Innovative Academy of Seed Design, Chinese Academy of Sciences, Wuhan, China

**\*For correspondence:**
zyin@ihb.ac.cn

[†]These authors contributed equally to this work.

**Competing interest:** The authors declare that no competing interests exist.

**Abstract** Disruption of androgen signaling is known to cause testicular malformation and defective spermatogenesis in zebrafish. However, knockout of *cyp17a1*, a key enzyme responsible for the androgen synthesis, in *ar-/-* male zebrafish paradoxically causes testicular hypertrophy and enhanced spermatogenesis. Because Cyp17a1 plays key roles in hydroxylation of pregnenolone and progesterone (P4), and converts 17α-hydroxypregnenolone to dehydroepiandrosterone and 17α-hydroxyprogesterone to androstenedione, we hypothesize that the unexpected phenotype in *cyp17a1-/-;androgen receptor* (*ar*)-/- zebrafish may be mediated through an augmentation of progestin/nuclear progestin receptor (nPgr) signaling. In support of this hypothesis, we show that knockout of *cyp17a1* leads to accumulation of 17α,20β-dihydroxy-4-pregnen-3-one (DHP) and P4. Further, administration of progestin, a synthetic DHP mimetic, is sufficient to rescue testicular development and spermatogenesis in *ar-/-* zebrafish, whereas knockout of *npgr* abolishes the rescue effect of *cyp17a1-/-* in the *cyp17a1-/-;ar-/-* double mutant. Analyses of the transcriptomes among the mutants with defective testicular organization and spermatogenesis (*ar-/-*, *ar-/-;npgr-/-* and *cyp17a-/-;ar-/-;npgr-/-*), those with normal phenotype (control and *cyp17a1-/-*), and rescued phenotype (*cyp17a1-/-;ar-/-*) reveal a common link between a downregulated expression of *insl3* and its related downstream genes in *cyp17a-/-;ar-/-;npgr-/-* zebrafish. Taken together, our data suggest that genetic or pharmacological augmentation of the progestin/nPgr pathway is sufficient to restore testis organization and spermatogenesis in zebrafish with the depletion of androgen signaling.

## Editor's evaluation

Deletion of the androgen receptor leads to testicular malformation and defective spermatogenesis in zebrafish. Paradoxically, simultaneous deletion of the androgen receptor and cyp17a1, a key

enzyme in the androgen synthesis pathway, leads to testicular hypertrophy and enhanced spermatogenesis. By analyzing a number of mutant zebrafish lines, the authors have provided new evidence suggesting that an elevation of progestin signaling can compensate for the loss of androgen receptor and progestin promotes testis organization and spermatogenesis via the nuclear progestin receptor.

## Introduction

Androgens and estrogens exert considerable influence on the formation of primary sex characteristic (PSC) and secondary sex characteristic (SSC) in fish. In the zebrafish model, the steroidogenic pathway in reproduction has been extensively investigated as zebrafish possess several advantages for research, including the ease of germ cells observation, gonadal dissection, fertility capacity assessment, knockout strategy (*Crowder et al., 2018*; *Lau et al., 2016*; *Li et al., 2020*; *Lu et al., 2017*; *Tang et al., 2018*; *Tang et al., 2016*; *Yin et al., 2017*; *Yu et al., 2018*; *Zhai et al., 2018*; *Zhu et al., 2015*), as well as chemical reagent administration, including estradiol (E2), testosterone (T), 11-ketotestosterone (11-KT), fadrozole, flutamide, and 17α,20β-dihydroxy-4-pregnen-3-one (DHP) (*Chen et al., 2013*; *Fenske and Segner, 2004*; *Shu et al., 2020*; *Zhai et al., 2018*; *Zhai et al., 2017*). Based on observations of *ar*- or *npgr*-deficient zebrafish, androgen signaling is reported to be essential for sex determination, testis organization, male-typical SSCs, and fertility (*Crowder et al., 2018*; *Li et al., 2020*; *Tang et al., 2018*; *Yu et al., 2018*; *Zhai et al., 2018*), while progestin signaling has been suggested to be important for female ovulation (*Tang et al., 2016*; *Wu and Zhu, 2020*).

In humans, the *CYP17A1* mutation causes pseudohermaphroditism, delay in sexual maturation, and absence of masculinization (*Kater and Biglieri, 1994*; *Marsh and Auchus, 2014*; *Yanase, 1995*; *Yanase et al., 1991*). In mice, *Cyp17a1* knockout leads to infertility and loss of sexual behaviors (*Liu et al., 2005*). In zebrafish, the knockout of *ar*, *cyp11c1* (encoding 11β-hydroxylase) or *cyp11a2* (encoding cytochrome P450 side-chain cleavage enzyme, the first and rate-limiting enzyme for steroid hormone biosynthesis) results in impaired spermatogenesis and disorganized testes, which is accompanied by androgen signaling insufficiency (*Crowder et al., 2018*; *Li et al., 2020*; *Oakes et al., 2020*; *Tang et al., 2018*; *Yu et al., 2018*; *Zhang et al., 2020*). However, as evidenced by a recent study reported from our laboratory, a phenotype of all-male *cyp17a1*-deficient fish that exhibited reductions of T and 11-KT in plasma and brain samples and loss of male-typical SSCs and mating behaviors with enhanced testicular development and spermatogenesis was observed (*Shu et al., 2020*; *Zhai et al., 2018*; *Zhai et al., 2017*). In another well-known freshwater fish model with an XX/XY genetic sex determination system, medaka, the *scl* mutant strain carrying a loss-of-function mutation in the *cyp17a1* gene also results in the loss of male-typical SSCs while spermatozoa develop normally (*Sato et al., 2008*). Similarly, results were also observed in another cyprinid fish, the common carp (*Cyprinus carpio* L.), with *cyp17a1* deletion (the *cyp17a1-/-* XX fish) developed normal testis structure with normal spermatogenesis and sperm capacity, and lost male-typical SSCs. Using artificial fertilization, the neomale common carp were mated with wild-type (WT) females (*cyp17a1+/+* XX genotype). We confirmed that all offspring from the neomale-WT female mating have the *cyp17a1±*XX genotype, and 100.00% normal development of gonads to ovaries (n > 500) (*Zhai et al., 2022*). However, other than impaired male-typical SSCs, the disorganized testicular development and impaired spermatogenesis were not observed in the *cyp17a1*-deficient cyprinid fish (zebrafish and common carp) and *scl* mutant medaka (*Sato et al., 2008*; *Zhai et al., 2022*; *Zhai et al., 2018*), but in tilapia (*Yang et al., 2021*). Our initial hypothesis was that, compared with the male-typical SSCs and mating behaviors, testicular development and spermatogenesis may not be susceptible to androgen signaling deficiency. Nevertheless, this hypothesis is not supported by the phenotypes of the *ar*, *cyp11a2*, and *cyp11c1* knockout zebrafish that show disorganized testes and impaired spermatogenesis. Therefore, an alternative pathway might exist in the *cyp17a1-/-* fish, facilitating testicular development and spermatogenesis when androgen signaling deteriorates.

Progestin signaling is important in mediating spermatogenesis, sperm maturation, and spermiation (*Baynes and Scott, 1985*; *Miura et al., 1992*; *Tubbs and Thomas, 2008*; *Vizziano et al., 1996b*). However, the *npgr* loss-of-function models diverged in phenotypes both in male mammals and fish. In mice, homozygous *progesterone receptor* (*Pr*) knockout males were fertile as WT and heterozygous male siblings (*Lydon et al., 1995*). Consistently, no evidence of fertility effects in the *npgr* knockout

male zebrafish has been reported (*Tang et al., 2016*; *Wu and Zhu, 2020*). In male tilapia, *npgr* knockout resulted in disorganized spermatogenic cysts, decreased sperm number, motility, spermatocytes, and spermatozoa, showing the essentiality for the fertility of XY tilapia (*Fang et al., 2018*). The underlying mechanism of this discrepancy has been only scarcely investigated. We hypothesize that the different requirements for progestin receptors and the alternative between progestin signaling and other pathways (or factors) may contribute to it.

Although an alternative pathway might exist in the *cyp17a1-/-* fish to facilitate testicular development and spermatogenesis when androgen signaling deteriorates, it had not yet been proposed. The *cyp17a1-/-* zebrafish generated in our laboratory shows stimulated testicular development and spermatogenesis with deteriorated androgen synthesis (*Zhai et al., 2018*). In the *cyp17a1-/-* fish, the accumulation of progestins, DHP, and P4, which are the upstream products of androstenedione and testosterone, was observed, possibly due to the loss of the critical 17α-hydroxylase and 17, 20-lyase activities of Cyp17a1 during gonadal steroidogenesis (*Tokarz et al., 2013*; *Zhai et al., 2018*). To dissect the roles of androgen signaling and progestin signaling in adult zebrafish, we established and analyzed a series of deficiency models of sex steroid signaling (*cyp17a1-/-*, *ar-/-*, *cyp17a1-/-;ar-/-*, *cyp17a1-/-;ar-/-;npgr-/-*, and *cyp17a1-/-;ar-/-;fshβ-/-*) and initiated a concurrent study of DHP administration in *ar-/-* males as well. Here, we report that an augmentation of progestin/nuclear progestin receptor (nPgr) signaling can sufficiently compensate for proper spermatogenesis and testis organization when androgen signaling deteriorates in the *cyp17a1-/-* and *cyp17a1-/-;ar-/-* zebrafish.

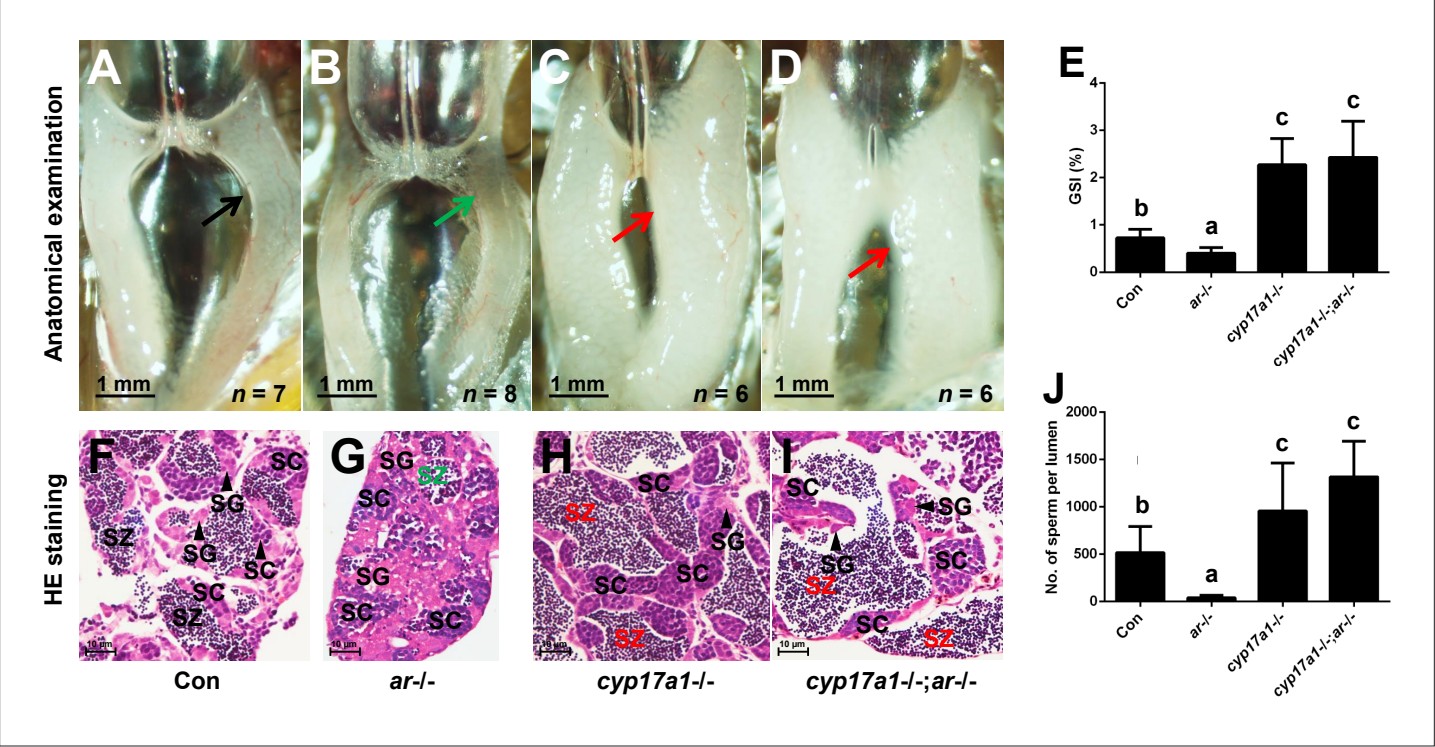

**Figure 1.** The alternative compensatory pathway induced by *cyp17a1* depletion is *ar*-independent. (**A–D**) Anatomical examination of the testes from control males, *ar-/-* males, *cyp17a1-/-* fish, and *cyp17a1-/-;ar-/-* fish at 6 months post-fertilization (mpf). Black and green arrows indicate the normal and impaired testis in control males and *ar-/-* males, respectively, whereas the red arrows indicate the hypertrophic testis in the *cyp17a1-/-* fish and *cyp17a1-/-;ar-/-* fish. (**E**) Gridpoint Statistical Interpolation (GSI) from the fish of the four genotypes at 6 mpf. (**F–I**) Histological analyses of the testes from control males, *ar-/-* males, *cyp17a1-/-* fish, and *cyp17a1-/-;ar-/-* fish at 6 mpf. Black and green letters of spermatozoa (SZ) indicate the normal and decreased number of SZ in control males and *ar-/-* males, respectively, whereas the red letters of SZ indicate the increased number of SZ in the *cyp17a1-/-* fish and *cyp17a1-/-;ar-/-* fish. (**J**) Statistical analysis of the SZ number in each lumen of seminiferous tubules from the fish of the four genotypes at 6 mpf. The letters in the bar charts (**E**) and (**J**) represent significant differences. SC: spermatocytes; SG: spermatogonia.

The online version of this article includes the following source data for figure 1:

**Source data 1.** GSI data (*Figure 1E*) and number of sperm per lumen (*Figure 1J*).

# Results

## *cyp17a1* knockout activates a compensatory pathway that promotes testis organization and spermatogenesis independent of Ar

In the *cyp17a1-/-* fish at 6 months post-fertilization (mpf), a significant increase in the number of spermatozoa and testicular development was observed (*Zhai et al., 2018*). We hypothesized that an alternative pathway facilitates testicular development and spermatogenesis in *cyp17a1-/-* fish. The *ar* mutation was introduced into the *cyp17a1-/-* fish to determine whether Ar contributes to the hypertrophic testis and the increase in the number of spermatozoa in the *cyp17a1-/-* fish. To generate the *cyp17a1-/-;ar-/-* fish, the double heterozygotes (*cyp17a1+/-;ar+/-*) among the F1 progeny were crossed. In agreement with previous studies, reduced testis size and defective spermatogenesis in the *ar-/-* male zebrafish were observed (*Crowder et al., 2018*; *Tang et al., 2018*; *Yu et al., 2018*; *Figure 1B and G*). However, anatomical examination (*Figure 1A–D*), Gridpoint Statistical Interpolation (GSI) statistical analysis (*Figure 1E*), histological analysis (*Figure 1F–I*), and spermatozoa number analysis (*Figure 1J*) demonstrated that, compared with that in the control males, the hypertrophic testis and increased spermatogenesis were observed in the *cyp17a1-/-;ar-/-* fish. These results support the hypothesis that *cyp17a1* knockout activates a compensatory pathway that promotes testis organization and spermatogenesis independent of Ar.

## DHP and P4 are accumulated in the *cyp17a1-/-* fish, not in *ar-/-* males

In our previous studies, we have demonstrated that the concentrations of T and 11-KT in both the plasma and brain were significantly decreased in the *cyp17a1-/-* fish at 3 mpf (*Shu et al., 2020*; *Zhai et al., 2018*; *Zhai et al., 2017*). Moreover, T and 11-KT restored the loss of male-typical anal fin coloration, breeding tubercles in the pectoral fin, and sexual behaviors when mating with females in the *cyp17a1-/-* fish (*Shu et al., 2020*). These results indicate that gonadal steroidogenesis is impaired in *cyp17a1-/-* fish. In the present study, the concentrations of testis T and 11-KT were evaluated in control males and *cyp17a1-/-* fish using ELISA. We observed that the testis T and 11-KT concentrations in the *cyp17a1-/-* fish at 3 mpf were significantly lower than that in the control males (*Figure 2—figure supplement 1A and B*). Subsequently, the whole-body contents of 11-KT, DHP, and P4 of the control

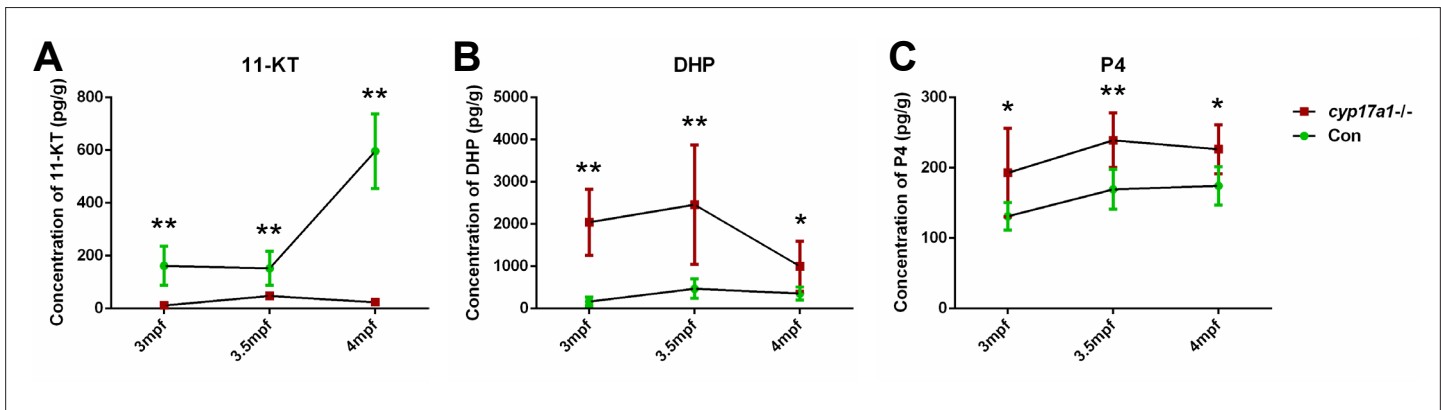

**Figure 2.** The 11-ketotestosterone (11-KT), 17α,20β-dihydroxy-4-pregnen-3-one (DHP), and progesterone (P4) measurements from whole-body lysates of the *cyp17a1-/-* fish and its control male siblings at 3 months post-fertilization (mpf), 3.5 mpf, and 4 mpf using ultra performance liquid chromatography tandem mass spectrometry (UPLC-MS/MS). (**A**) 11-KT, (**B**) DHP, and (**C**) P4. *p<0.05; **p<0.01.

The online version of this article includes the following source data and figure supplement(s) for figure 2:

**Source data 1.** 11-KT, DHP and P4 concentrations measured from whole-body lysate of the cyp17a1-/- fish at respective stages.

**Figure supplement 1.** Depletion of *cyp17a1* resulted in decreased concentrations of testosterone (T) and 11-ketotestosterone (11-KT) in the testes as measured by ELISA.

**Figure supplement 1—source data 1.** *Figure 2—figure supplement 1* for T and 11-KT concentrations.

**Figure supplement 2.** The 11-ketotestosterone (11-KT), 17α,20β-dihydroxy-4-pregnen-3-one (DHP), and progesterone (P4) measurements from whole-body lysates of the *ar-/-* males and their control male siblings at 3 months post-fertilization (mpf) using ultra performance liquid chromatography tandem mass spectrometry (UPLC-MS/MS).

**Figure supplement 2—source data 1.** Source data for 11-KT, DHP and P4 concentrations measured.

males and *cyp17a1-/-* fish at 3, 3.5, and 4 mpf were measured using ultra performance liquid chromatography tandem mass spectrometry (UPLC-MS/MS). Compared with their control male siblings at their corresponding stages, a significant decrease in 11-KT and an increase in DHP and P4 were observed in the *cyp17a1-/-* fish (*Figure 2A–C*). The whole-body contents of 11-KT, DHP, and P4 of the control males and *ar-/-* males at 3 mpf were also compared, which did not show any significant difference (*Figure 2—figure supplement 2A–C*). The phenomena of the increases in DHP and P4 and decrease in 11-KT in the whole-body lysates of *cyp17a1-/-* fish detected using UPLC-MS/MS were identical to that in the plasma or testes of *cyp17a1-/-* fish detected using ELISA reported previously (*Zhai et al., 2018*). These results not only confirmed that androgen biosynthesis is impaired in the *cyp17a1-/-* fish, but also reminded us that compared with the *cyp17a1-/-* males that have higher concentrations of DHP and P4 than their control male siblings, the *ar-/-* males have comparable concentrations of DHP and P4 with their control male siblings.

## Progestin signaling is sufficient to compensate the androgen signaling insufficiency

To confirm whether the increased concentration of DHP is responsible for testis organization and spermatogenesis in the *cyp17a1-/-* fish, *ar-/-* males were treated with 0.067 µg/mL DHP from 50 to 90 days post-fertilization (dpf). The results of anatomical examination (*Figure 3A–D*), GSI statistical analysis (*Figure 3E*), histological analyses (*Figure 3F–I*), and spermatozoa number analysis (*Figure 3J*) showed that DHP was sufficient to rescue the phenotypes of testis organization and spermatogenesis in *ar-/-* males (*Figure 3D, E, I, and J*). The effective restoration of testis organization and spermatogenesis after DHP treatment in *ar-/-* males confirmed that the accumulated DHP might facilitate testicular development and spermatogenesis by compensating for the efficiency of androgen signaling in the *cyp17a1-/-* fish. The *ar-/-* males after DHP administration showed a more than threefold higher concentration of DHP compared with those reared in the system water (*ar-/-* males reared in system water: 166.1 ± 70.46, n = 5; *ar-/-* males reared in DHP: 578.8 ± 379.6, n = 5) (*Figure 3K*). However, the fertility (sperm capacity) of *ar-/-* males was not rescued by DHP treatment as the fertilization ratios of *ar+/+* males and *ar-/-* males after the DHP treatment were both significantly downregulated compared with that of *ar+/+* males reared in the system water when artificial fecundation was performed with WT females (*Figure 3—figure supplement 1*).

## The augmentation of progestin pathway facilitates testis organization, and spermatogenesis in the *cyp17a1-/-;ar-/-* fish depends on nPgr

The significantly elevated concentration of DHP in the *cyp17a1-/-* fish and the effective restoration of testis organization and spermatogenesis of *ar-/-* males after DHP treatment reinforce the fact that the in vivo progestin signaling compensates for androgen signaling insufficiency (*Figures 2 and 3*). Therefore, the *npgr* mutation was introduced into the *cyp17a1-/-;ar-/-* fish. To generate the *cyp17a1-/-;ar-/-;npgr-/-* fish, the triple heterozygotes (*cyp17a1+/-;ar+/-;npgr+/-*) among the F1 progeny were crossed. The fish of the eight genotypes from the *cyp17a1+/-;ar+/-;npgr+/-* offspring, including *cyp17a1+/+;ar+/+;npgr+/+* males (control males), *ar-/-* males, *cyp17a1-/-* fish, *npgr-/-* males, *cyp17a1-/-;ar-/-* fish, *cyp17a1-/-;npgr-/-* fish, *ar-/-;npgr-/-* males, and *cyp17a1-/-;ar-/-;npgr-/-* fish were analyzed. Compared with that in the *cyp17a1-/-* fish and *cyp17a1-/-;ar-/-* fish, the results of the anatomical examination (*Figure 4A–H*), GSI statistical analysis (*Figure 4I*), histological analyses (*Figure 4J–Q*), and spermatozoa number analysis (*Figure 4R*) demonstrated that the GSI and spermatozoa number were both significantly decreased in the *cyp17a1-/-;npgr-/-* fish (*Figure 4I and R*). These results suggest that a potential compensatory role of progestin signaling exists in the *cyp17a1-/-* fish. On the other hand, compared with that in the *cyp17a1-/-;ar-/-* fish, testis organization and spermatogenesis failed in the *cyp17a1-/-;ar-/-;npgr-/-* fish (*Figure 4H and Q*), which showed a similar pattern in the *ar-/-* males and *ar-/-;npgr-/-* males (*Figure 4B, G, K and P*). In addition, compared with that in the *cyp17a1-/-* and *cyp17a1-/-;ar-/-* fish, greater spermatogonia and spermatocytes, and less spermatozoa in the *ar-/-* males, *ar-/-;npgr-/-* males, and *cyp17a1-/-;ar-/-;npgr-/-* fish were observed in the histological analyses of testes (*Figure 4J–Q*). These results suggest that the compensatory pathway induced by *cyp17a1* knockout to promote testis organization and spermatogenesis exists and depends on nPgr.

The germ cell marker, Vasa, was examined by immunofluorescence staining in *cyp17a1+/+;ar+/+;npgr+/+* males (control males), *ar-/-* males, *cyp17a1-/-* fish, *cyp17a1-/-;ar-/-* fish,

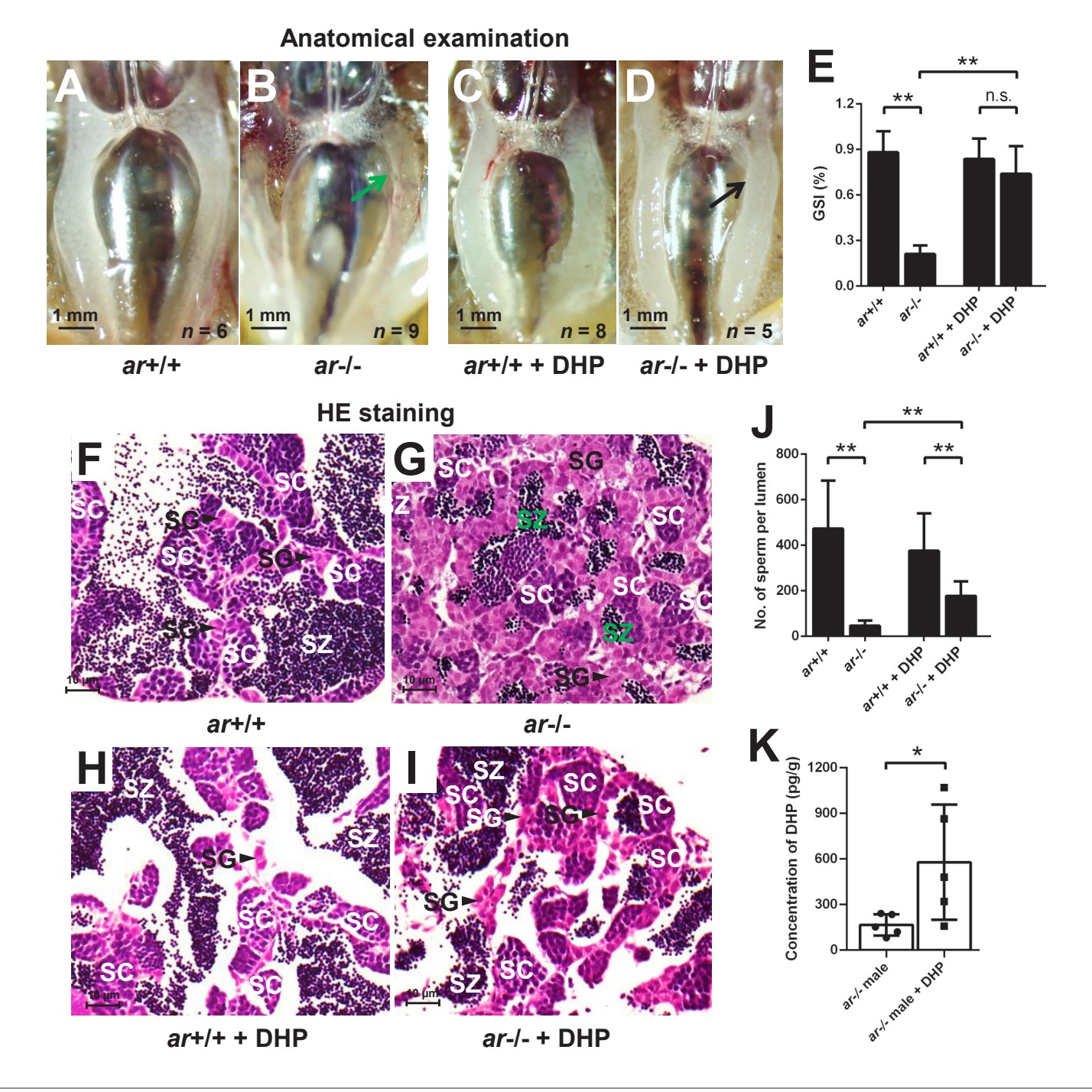

**Figure 3.** 17α,20β-Dihydroxy-4-pregnen-3-one (DHP) treatment rescues the testis organization and spermatogenesis in the *ar-/-* males. (**A, B**) Anatomical examination of *ar+/+* males and *ar-/-* males reared in system water. (**C, D**) Anatomical examination of *ar+/+* males and *ar-/-* males reared in DHP. Green and black arrows indicate the decreased and normal testis, respectively, in the *ar-/-* males. (**E**) Gridpoint Statistical Interpolation (GSI) from *ar+/+* males and *ar-/-* males reared in system water and DHP, respectively. (**F, G**) Histological analyses of testes from *ar+/+* males and *ar-/-* males reared in system water. (**H, I**) Histological analyses of testes from *ar+/+* males and *ar-/-* males reared in DHP. (**J**) Statistical analysis of the spermatozoa (SZ) number in each lumen of seminiferous tubules from the *ar+/+* males and *ar-/-* males reared in system water and DHP, respectively. (**K**) The DHP concentrations measurement of the *ar-/-* males reared in system water and DHP, respectively. *p<0.05; **p<0.01. n.s., no significance; SC: spermatocytes; SG: spermatogonia.

The online version of this article includes the following source data and figure supplement(s) for figure 3:

*Figure 3 continued on next page*

*Figure 3 continued*

**Source data 1.** GSI value for *Figure 3E*; number of sperm per lumen for *Figure 3J*; DHP concentrations for *Figure 3K*.

**Figure supplement 1.** The fertilization ratio analysis.

**Figure supplement 1—source data 1.** The fertilization ratios meatured of wildtype males and ar-/- males after DHP treatments.

and *cyp17a1-/-;ar-/-;npgr-/-* fish. As reported previously, Vasa signal intensity enriched in the cytoplasm decreases with the differentiation of germ cells from spermatogonia to advanced stages of spermatids and sperm (*Dai et al., 2021*; *Yu et al., 2018*). Compared with that in the control males (*Figure 5A–C*), the results of Vasa immunofluorescence localization showed a reduced number of spermatozoa in the lumen of seminiferous tubules of *ar-/-* males (*Figure 5D–F*), and an increased number of spermatozoa in the *cyp17a1-/-* fish (*Figure 5G–I*) and *cyp17a1-/-;ar-/-* fish (*Figure 5J–L*), again supporting that the reduced number of spermatozoa in the lumen of seminiferous tubules of *ar-/-* males was rescued after the *cyp17a1* knockout (in the *cyp17a1-/-;ar-/-* fish). However, the restoration failed in the *cyp17a1-/-;ar-/-* fish with deletion of *npgr* (in the *cyp17a1-/-;ar-/-;npgr-/-* fish) (*Figure 5M–O*), which showed greater spermatogonia, primary spermatocytes and secondary spermatocytes, and less spermatozoa filling in the lumen of seminiferous tubules of *cyp17a1-/-;ar-/-;npgr-/-* fish (*Figure 5O*) and the *ar-/-* males (*Figure 5F*). These results suggest that spermatogenesis in the *ar-/-* males and *cyp17a1-/-;ar-/-;npgr-/-* fish were impaired during the second meiosis phase.

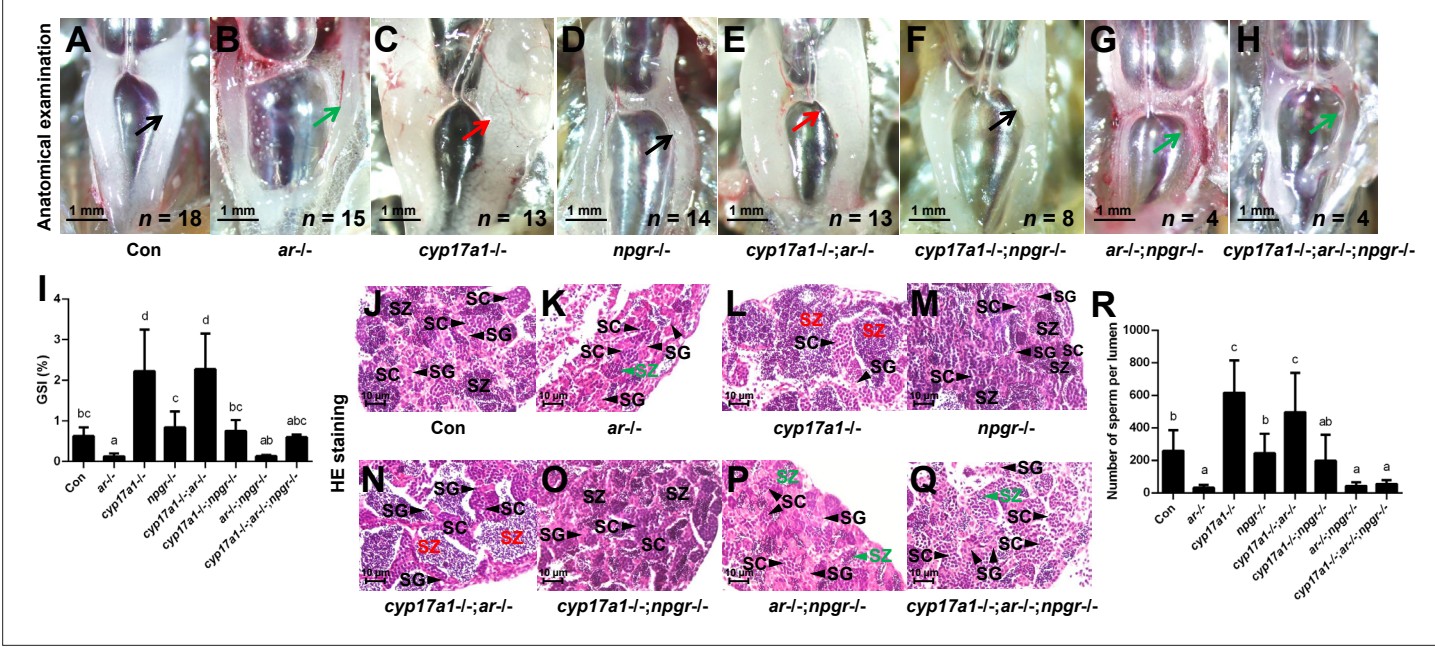

**Figure 4.** The alternative compensatory pathway induced by *cyp17a1* depletion is *npgr*-dependent. (**A–H**) Anatomical examination of the testes of control males, *ar-/-* males, *cyp17a1-/-* fish, *npgr-/-* males, *cyp17a1-/-;ar-/-* fish, *cyp17a1-/-;npgr-/-* fish, *ar-/-;npgr-/-* males, and *cyp17a1-/-;ar-/-;npgr-/-* fish at 6 months post-fertilization (mpf). Black and green arrows indicate the normal and impaired testis in control males, *ar-/-* males, *npgr-/-* males, *cyp17a1-/-;npgr-/-* fish, *ar-/-;npgr-/-* males, and *cyp17a1-/-;ar-/-;npgr-/-* fish, respectively, whereas the red arrows indicate the hypertrophic testis in the *cyp17a1-/-* fish and *cyp17a1-/-;ar-/-* fish. (**I**) Gridpoint Statistical Interpolation (GSI) from the fish of the eight genotypes at 6 mpf. (**J–Q**) Histological analyses of the testes from the fish of the eight genotypes at 6 mpf. Black and green letters indicate the normal and decreased number of spermatozoa (SZ) in control males, *ar-/-* males, *npgr-/-* males, *cyp17a1-/-;npgr-/-* fish, *ar-/-;npgr-/-* males, and *cyp17a1-/-;ar-/-;npgr-/-* fish, respectively, whereas the red letters of SZ indicate the increased number of SZ in the *cyp17a1-/-* fish and *cyp17a1-/-;ar-/-* fish. (**R**) Statistical analysis of the SZ number in each lumen of seminiferous tubules from the fish of the eight genotypes at 6 mpf. The letters in the bar charts (**I**) and (**R**) represent significant differences. SC: spermatocytes; SG: spermatogonia.

The online version of this article includes the following source data for figure 4:

**Source data 1.** GSI values for *Figure 4I*; number of sperm per lumen for *Figure 4R*.

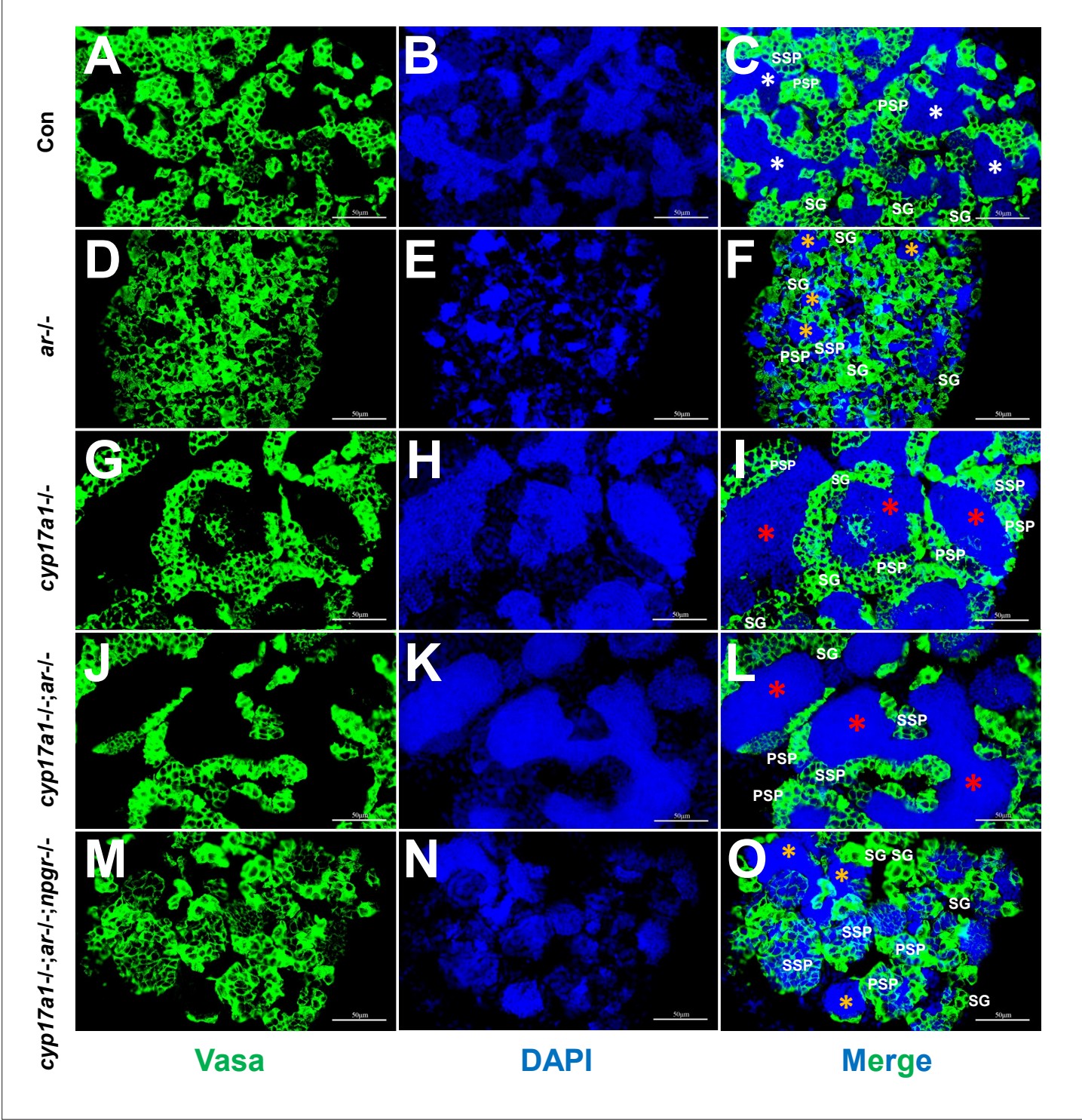

**Figure 5.** Germ cells were visualized by immunofluorescence staining of Vasa. (**A–C**) Control male. (**D–F**) *ar-/-* male. (**G–I**) *cyp17a1-/-* fish. (**J–L**) *cyp17a1-/-;ar-/-* fish. (**M–O**) *cyp17a1-/-;ar-/-;npgr-/-* fish. Nuclear DNA was stained with 4',6-diamidino-2-phenylindole (DAPI). White and yellow asterisks in panels (**C**), (**F**), and (**O**) indicate the normal and decreased number of spermatozoa (SZ), respectively, whereas the red asterisks in panels (**I**) and (**L**) indicate the increased number of SZ in each lumen of seminiferous tubule. SG: spermatogonia; PSP: primary spermatocyte; SSP: secondary spermatocyte.

## Abnormal expression of germ cell differentiation-related genes in the *cyp17a1-/-;ar-/-;npgr-/-* fish

Comparisons of transcriptome in the testes of the control males, *ar-/-* males, *cyp17a1-/-* fish, *npgr-/-* males, *cyp17a1-/-;ar-/-* fish, *cyp17a1-/-;npgr-/-* fish, *ar-/-;npgr-/-* males, and *cyp17a1-/-;ar-/-;npgr-/-* fish, were performed. For the markers of Leydig cells, the genes related to gonadal steroidogenesis, including *star*, *hsd3β1*, *cyp11a2*, and *cyp11c1*, were all increased in the testes of *ar-/-* males, *npgr-/-* males, *cyp17a1-/-;ar-/-* fish, *cyp17a1-/-;npgr-/-* fish, *ar-/-;npgr-/-* males, and *cyp17a1-/-;ar-/-;npgr-/-* fish. Interestingly, the expression levels of *insl3*, another specific gene of the Leydig cells, were significantly decreased in the *ar-/-* males, *ar-/-;npgr-/-* males, and *cyp17a1-/-;ar-/-;npgr-/-* fish compared with that in the controls (*Figure 6A*). For the markers of Sertoli cells, the expression levels of *sox9a*, *amh*, *dmrt1*, *igf3*, and *rxfp2b* (the receptor of Insl3) were all decreased in the *ar-/-* males, *cyp17a1-/-* fish, *npgr-/-* males, *cyp17a1-/-;ar-/-* fish, *cyp17a1-/-;npgr-/-* fish, *ar-/-;npgr-/-* males, and *cyp17a1-/-;ar-/-;npgr-/-* fish compared with that in the controls (*Figure 6B*). For the markers of the germ cells, the expression levels of *dazl*, *vasa*, *dnd*, *nanos1*, *nanos2*, *piwil1*, and *piwil2* were all upregulated in the *cyp17a1-/-;ar-/-;npgr-/-* fish compared with that in the controls, with divergent transcriptional expression patterns in the testes of fish with other genotypes. Notably, *rxfp2a* (another receptor of Insl3) and the retinoic acid-degrading enzyme *cyp26a1* were downregulated and upregulated, respectively, in the *cyp17a1-/-;ar-/-;npgr-/-* fish compared with that in the control males (*Figure 6C and D*). The expression levels of *hsd3β1*, *insl3*, *igf3*, *dnd*, and *cyp26a1* were selected and further verified by qPCR (*Figure 6E–I*).

Subsequently, we analyzed the expression of genes in fish of different genotypes. Since the normal spermatogenesis has been observed in the control males, *cyp17a1-/-;ar-/-* fish, and *cyp17a1-/-* fish, while the defective spermatogenesis has been observed in *cyp17a1-/-;ar-/-;npgr-/-* fish. Therefore, the expressed genes in *cyp17a1-/-;ar-/-;npgr-/-* fish compared with that in the control males, *cyp17a1-/-;ar-/-* fish, and *cyp17a1-/-* fish, respectively, were analyzed and presented in a Venn diagram (*Figure 6—figure supplement 1A*), which could be used to identify the common transcripts responsible for the normal spermatogenesis course. Out of 1380 annotated genes, we identified a total of 148 differentially expressed genes in the overlapped region, such as the downregulated *gonadal somatic cell-derived factor* (*gsdf*), *npgr*, *axonemal dynein assembly factor 3* (*dnaaf3*), *insl3,* and upregulated *inhibin subunit beta B* (*inhbb*) (*Supplementary file 1*). The downregulated *npgr* may have resulted from the *npgr* deletion-mediated premature mRNA decay in the *cyp17a1-/-;ar-/-;npgr-/-* fish (*El-Brolosy et al., 2019*), and the aberrant expression of *gsdf*, *insl3,* and *inhbb* may contribute to the compromised testis organization and spermatogenesis in the *cyp17a1-/-;ar-/-;npgr-/-* fish.

On the other hand, the defective spermatogenesis has been observed in the *ar-/-* males, *ar-/-;npgr-/-* males, and *cyp17a1-/-;ar-/-;npgr-/-* fish. Therefore, the expressed genes in *ar-/-* males, *ar-/-;npgr-/-* males, and *cyp17a1-/-;ar-/-;npgr-/-* fish compared with that in the control males, respectively, were analyzed and summarized in *Figure 6—figure supplement 1B*, which can help to determine the common transcriptional changes responsible for the defective spermatogenesis. Out of 1315 annotated genes, we identified a total of 111 differentially expressed genes in the overlapped region, such as the downregulated *axonemal central pair apparatus protein* (*hydin*), *RNA binding motif protein 47* (*rbm47*), *axonemal dynein assembly factor 1* (*dnaaf1*), *outer dense fiber of sperm tails 3B* (*odf3b*), and *insl3* (*Supplementary file 1*). These results demonstrated that phenotypes in the *ar-/-* males, *ar-/-;npgr-/-* males, and *cyp17a1-/-;ar-/-;npgr-/-* fish may be caused by the dysregulated expressions of genes involved in the process of spermatogenesis and structure organization of sperm.

## Discussion

Spermatogenesis is a highly coordinated developmental process involving diploid spermatogonia proliferation and differentiation, during which diploid spermatogonial stem cells produce a large number of highly differentiated spermatozoa (*Schulz et al., 2010*). Progestin signaling, which mainly acts via the progestin receptor expressed in Sertoli cells, is closely related to spermatogenesis (*Schulz et al., 2010*). First, non-flagellated germ cells in rainbow trout testes can synthesize DHP in the early phase of spermatogenesis (*Vizziano et al., 1996a*). In salmonid fish, the plasma concentration of DHP peaks during the progression of spermatogonial proliferation (*Scott and Sumpter, 1989*). Second, DHP treatment induces spermiation in *Salmonidae* and *Cyprinidae* (*Ueda et al., 1985*), increases

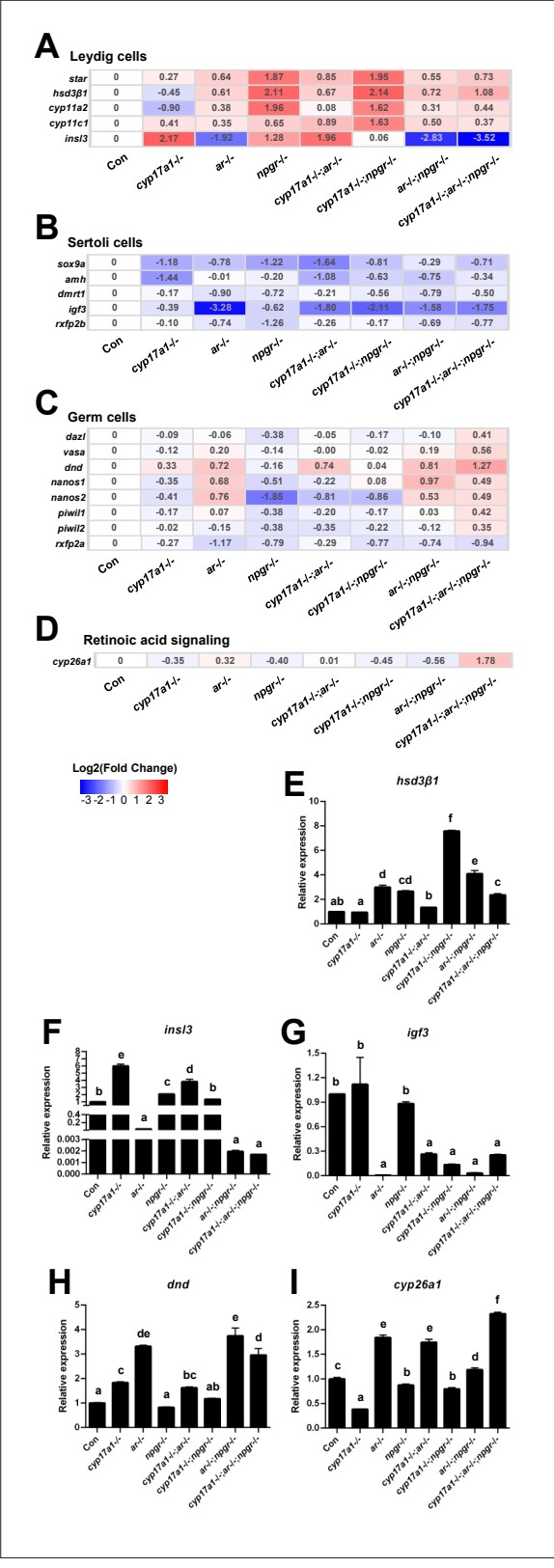

**Figure 6.** Gene expression analyses of testis Leydig cells, Sertoli cells, and germ cells. (**A–D**) Heatmap of the candidate genes. (**A**) The heatmap of *star*, *hsd3β1*, *cyp11a2*, *cyp11c1*, and *insl3* of Leydig cells. (**B**) The heatmap of *sox9a*, *amh*, *dmrt1*, *igf3*, and *rxfp2b* of Sertoli cells. (**C**) The heatmap of *dazl*, *vasa*, *dnd*, *nanos1*, *nanos2*, *piwil1*, *piwil2*, and *rxfp2a* (another receptor of Insl3) of germ cells. (**D**) The expression of the retinoic acid-degrading

*Figure 6 continued on next page*

*Figure 6 continued*

enzyme, *cyp26a1*. (**E–H**) The expression of the selected genes with qPCR for confirmation of transcriptome analyses. (**E**) *hsd3β1*. (**F**) *insl3*. (**G**) *igf3*. (**H**) *dnd*. (**I**) *cyp26a1*. The letters in the bar charts represent significant differences.

The online version of this article includes the following source data and figure supplement(s) for figure 6:

**Source data 1.** Real-time RT PCR results for *Figure 6E-I*.

**Figure supplement 1.** Venn diagram of gene expression profiles in testes.

**Figure supplement 2.** The upregulated *fshβ* contributes to the hypertrophic testis and enhanced spermatogenesis in the *cyp17a1-/-* fish and *cyp17a1-/-;ar-/-* fish at 6 months post-fertilization (mpf).

**Figure supplement 2—source data 1.** The GSI values for *Figure 6—figure supplement 2F*.

**Figure supplement 3.** Expression analysis of pituitary *fshβ* in the control males, *cyp17a1-/-* fish, *ar-/-* males, *npgr-/-* males, *cyp17a1-/-;ar-/-* fish, *cyp17a1-/-;npgr-/-* fish, *ar-/-;npgr-/-* males, and *cyp17a1-/-;ar-/-;npgr-/-* fish at 6 months post-fertilization (mpf). The letters in the bar chart represent significant differences.

**Figure supplement 3—source data 1.** Real-time PCR results of pituitary fshbeta transcriptional levels in zebrafish of various genotypes.

---

milt production (*Baynes and Scott, 1985*), and regulates meiosis in male eels (*Miura et al., 2006*). In the testis of Nile tilapia, *npgr* was expressed in Sertoli cells surrounding spermatogonia and spermatids. The treatment with RU486, a synthetic nPgr antagonist, inhibits progestin signaling and disrupts spermatogenesis in Nile tilapia. The expression levels of *piwil1*, *dazl*, and *sycp3* genes were found to be downregulated in the testes of RU486-treated fish (*Liu et al., 2014*). The GSI and the expression levels of the germ cell markers *piwil1*, *dazl*, and the meiotic marker, *sycp3*, increased after 2-week E2 exposure in the presence of DHP (*Chen et al., 2013*). Third, *npgr*-knockout male tilapia exhibited disorganized spermatogenic cysts, decreased sperm number and motility, as well as spermatocytes and spermatozoa (*Fang et al., 2018*). Although progestin signaling may play important roles in spermatogenesis, the in vivo evidence and its relationship with androgen signaling are yet to be elucidated, especially in the species that the males with *npgr* mutation did not exhibit obvious phenotypes in reproduction.

To our knowledge, the *cyp17a1-/-* zebrafish generated by us is a unique model exhibiting deteriorated androgen signaling but enhanced testicular development and spermatogenesis (*Li et al., 2020*; *Zhai et al., 2018*). In our newly generated *cyp17a1-/-;ar-/-* fish, the phenotype with testicular development and spermatogenesis similar to *cyp17a1-/-* fish was observed. The introduction of *cyp17a1* knockout successfully rescued the impaired testis organization and spermatogenesis in *ar-/-* males, clearly demonstrating the existence of an alternative androgen-independent signaling pathway promoting testis organization and spermatogenesis in the *cyp17a1-/-* fish (*Figure 1*). Zebrafish Ar showed an affinity for the non-androgenic steroid progesterone in the high nanomolar range (*de Waal et al., 2008*). The abnormalities in the *ar-/-* male zebrafish rescued by a compensatory mechanism demonstrated that the restorations of the testicular development and spermatogenesis after the introduction of the *cyp17a1*-deletion are not dependent on Ar (*Figure 1*). DHP and P4 are the androgen upstream products and accumulate in the *cyp17a1-/-* fish as measured using ELISA (*Zhai et al., 2018*) and UPLC-MS/MS (*Figure 2*). DHP is a fish-specific progestin that might play critical roles in spermatogenesis, sperm maturation, and spermiation (*Fang et al., 2018*). The administration of DHP effectively restored the phenotypes of GSI and the number of spermatozoa in *ar-/-* males (*Figure 3*). In addition, compared with the *cyp17a1-/-;ar-/-* fish that showed the hypertrophic testis and enhanced spermatogenesis, the *cyp17a1-/-;ar-/-;npgr-/-* fish exhibited phenotypes of defective spermatogenesis and testis organization, which are also seen in the *ar-*, *cyp11a2-*, or *cyp11c1*-deficient male zebrafish (*Li et al., 2020*; *Oakes et al., 2020*; *Tang et al., 2018*; *Yu et al., 2018*). These suggest that the accumulated progestin may play an alternative signaling role in promoting testis organization and spermatogenesis, which is independent of androgen signaling (in *cyp17a1-/-* fish and *cyp17a1-/-;ar-/-* fish) (*Figures 4 and 5*).

In mice with knockout of *Pr*, a member of the nuclear receptor superfamily of transcription factors, homozygous females displayed defects in reproductive tissues. These defects included the inability to ovulate, uterine hyperplasia and inflammation, severely limited mammary gland development, and loss of stereotypical sexual behavior (*Lydon et al., 1996*). In fact, larger testes and

greater sperm production have been observed in male *Pr-/-* mice (*Lue et al., 2013*; *Schneider et al., 2005*). Combined with the observations in the *cyp17a1-/-* fish and *Pr-/-* male mice, we speculate that the stimulated effects on spermatogenesis in the testis are not caused by the depleted actions of progestin or androgen signaling. These effects are likely to stem from the highly coordinatively regulated actions of androgen and progestin signaling cascades. In teleosts, various effects of progestins, including oocyte maturation, ovulation, spermatogenesis initiation, and spermatogenesis stimulation, have been previously reported (*Chen et al., 2013*; *Miura et al., 2006*; *Nagahama and Yamashita, 2008*). However, most early studies on teleosts have been conducted through progestin administration without robust genetic studies. Considering this evidence, the unimpaired fertility observed in the *npgr* homozygous males may be attributed to the presence of androgen signaling (*Tang et al., 2016*; *Zhu et al., 2015*). Recently, subfertility has been observed in *npgr*-deficient male tilapia (*Oreochromis niloticus*), which provides the first genetic evidence of the functions of progestin signaling in teleost spermatogenesis (*Fang et al., 2018*). The decreased levels of pituitary *follicle-stimulating hormone subunit β* (*fshβ*) and testis *follicle stimulating hormone receptor* (*fshr*) in the *npgr*-deficient tilapia were attributed to impaired spermatogenesis, the decline of milting, and livability of offspring (*Fang et al., 2018*). Nevertheless, the *npgr*-deficient male tilapia can successfully sire offspring, which might be due to the presence of androgen signaling, as evidenced by the increased concentration of 11-KT (*Fang et al., 2018*). On the other hand, the *CYP17A1* deficiency in humans and mice results in disorders of sex development and absence of masculinization (*Kater and Biglieri, 1994*; *Liu et al., 2005*; *Marsh and Auchus, 2014*; *New, 2003*; *Yanase, 1995*; *Yanase et al., 1991*), as well as accumulated P4, 11-deoxycorticosterone, and corticosterone (*Auchus, 2017*). We believe that the role of progestin signaling in facilitating spermatogenesis and testis organization to compensate for androgen signaling insufficiency is highly divergent between fish and mammals. Unfortunately, the DHP content in people with 17-hydroxylase/17,20-lyase deficiency was not measured and analyzed in that previous study (*Auchus, 2017*).

It has been reported that the activation of the gonadal-pituitary feedback axis was observed in *cyp17a1* and *cyp19a1a* knockout fish, as evidenced by the significant upregulation of pituitary *fshβ* and *lhβ* (*Li et al., 2020*; *Tang et al., 2017*; *Zhai et al., 2018*). These mutants also displayed impaired sex steroid biosynthesis, as measured by sex steroid concentrations in homozygous fish (*Li et al., 2020*; *Tang et al., 2017*; *Zhai et al., 2018*). The upregulation of gonadotropins observed in these knockout zebrafish could be attributed to the absence of negative feedback due to androgen or estrogen deficiency (*Chen et al., 2017*; *Tang et al., 2017*; *Zhai et al., 2018*). In the testes of adult zebrafish, Fshβ stimulates the proliferation and differentiation of spermatogonia and its entry into meiosis (*Holdcraft and Braun, 2004*; *Nobrega et al., 2015*; *Patiño et al., 2001*; *Ramaswamy and Weinbauer, 2014*). This supports the observations in the *cyp17a1-/-* fish and *cyp19a1a-/-* fish, as well as our newly generated *cyp17a1-/-;ar-/-* fish, in which testicular development and spermatogenesis were stimulated (*Figure 6—figure supplement 2D, F and J*). Compared with that in the *cyp17a1-/-;ar-/-* fish, the normal GSI and spermatozoa number displayed in *cyp17a1-/-;ar-/-;fshβ-/-* fish (*Figure 6—figure supplement 2E, F and K*) also support that the upregulated Fshβ contributes to the hypertrophic testicular development and overactivated spermatogenesis in the *cyp17a1-/-;ar-/-* zebrafish. By monitoring the expression of pituitary *fshβ* in fish of the aforementioned eight genotypes, we found that the expression of *fshβ* was upregulated in the *cyp17a1-/-* fish and *cyp17a1-/-;ar-/-* fish. This was an expected result based on our previous study (*Zhai et al., 2018*). The *npgr* depletion did not affect pituitary *fshβ* expression as *npgr-/-* males and *ar-/-;npgr-/-* males exhibited comparable *fshβ* expression to the control males. However, the additional deletion of *cyp17a1* significantly upregulated *fshβ* expression in the males of these genotypes (in the *cyp17a1-/-;npgr-/-* fish and *cyp17a1-/-;ar-/-;npgr-/-* fish) (*Figure 6—figure supplement 3*). These results demonstrated that *cyp17a1-/-;ar-/-;npgr-/-* fish have impaired spermatogenesis and testis organization, and upregulated pituitary *fshβ*.

The upregulations of gonadal steroidogenesis-related genes, *star*, *hsd3β1*, *cyp11a2*, and *cyp11c1*, in Leydig cells were observed in the *ar-/-* males, *npgr-/-* males, *cyp17a1-/-;ar-/-* fish, *cyp17a1-/-;npgr-/-* fish, *ar-/-;npgr-/-* males, and *cyp17a1-/-;ar-/-;npgr-/-* fish. The upregulated expression of steroidogenesis-related genes in the *ar-/-* males has been reported previously (*Tang et al., 2018*), which may be attributed to the positive feedback effect caused by androgen or progestin signaling insufficiency. Another marker of the Leydig cells, *insl3*, which has been reported to be essential for maintaining germ cell differentiation, was significantly decreased in the *ar-/-* males, *ar-/-;npgr-/-* males,

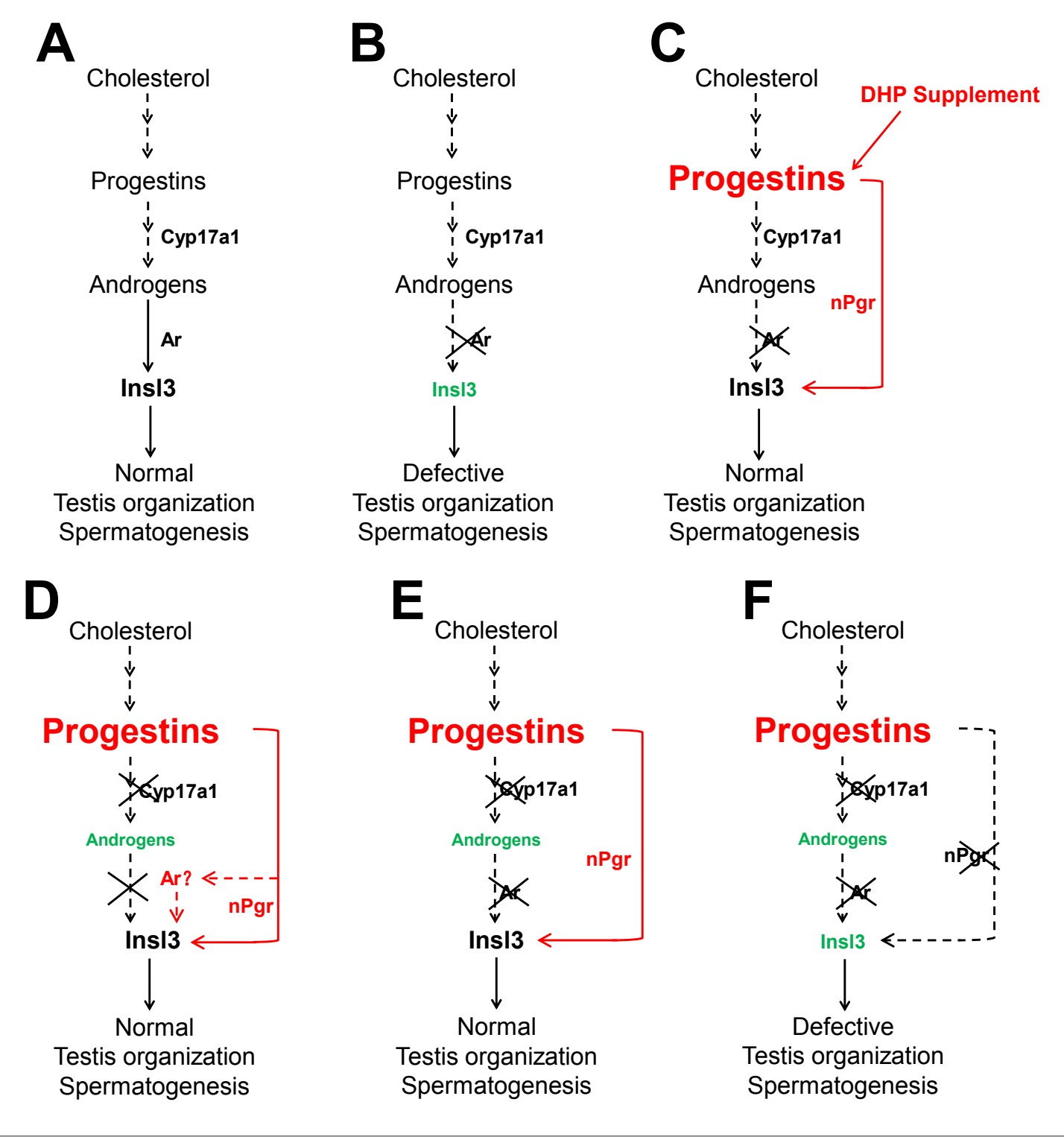

**Figure 7.** The potential regulatory network of androgens and progestins regulating testis organization and spermatogenesis via Insl3. (**A**) Androgen signaling is essential in promoting testis organization and spermatogenesis in the control males. (**B**) Testis organization and spermatogenesis is impaired in the *ar-/-* males. (**C**) The impaired testis organization and spermatogenesis in the *ar-/-* males could be rescued by 17α,20β-dihydroxy-4-pregnen-3-one (DHP) supplement. (**D**) Testis organization and spermatogenesis proceeded normally in the *cy17a1-/-* fish, resulting from the enhanced progestin signaling caused by *cyp17a1* depletion. (**E**) The alternative compensatory pathway induced by *cyp17a1* depletion is *ar*-independent. (**F**) The alternative compensatory pathway induced by *cyp17a1* depletion is *npgr*-dependent, demonstrating a high progestin/nuclear progestin receptor (nPgr) signaling pathway in promoting testis organization and spermatogenesis independent of androgen signaling. Red letters and lines indicate the upregulation of

*Figure 7 continued on next page*

*Figure 7 continued*

the progestin/nPgr signaling pathway, while green letters indicate the decreased concentration of androgens or downregulated *insl3*. The dotted lines indicate the brief description with omission or the potential existence of the proposed model.

and *cyp17a1-/-;ar-/-;npgr-/-* fish (the lowest), the only three genotype groups examined with impaired spermatogenesis and testis organization (**Figure 6A**). Considering the downregulation of *insl3* expression shown in the Venn diagram (**Figure 6—figure supplement 1A and B**), and reported in both *cyp11c1* and *cyp11a2* mutant male zebrafish, which are associated with disorganized testicular structure and significantly decreased numbers of mature spermatozoa (**Li et al., 2020**; **Zhang et al., 2020**), it is reasonable to speculate that *insl3* may be a target that is co-regulated by androgen signaling and high level of progestin signaling. Among the upstream signals of the *insl3*, the role of the accumulated progestin may be a compensatory signaling pathway that regulates testis organization and spermatogenesis in the absence of androgen signaling. The expression levels of the markers of Sertoli cells, *sox9a*, *amh*, *dmrt1*, and *igf3* were decreased in the *ar-/-* males, *cyp17a1-/-* fish, *npgr-/-* males, *cyp17a1-/-;ar-/-* fish, *cyp17a1-/-;npgr-/-* fish, *ar-/-;npgr-/-* males, and *cyp17a1-/-;ar-/-;npgr-/-* fish (**Figure 6B**), indicating impaired functions of Sertoli cells in these fish. The divergent expression patterns of the germ cell markers, *dazl*, *vasa*, *dnd*, *nanos1*, *nanos2*, *piwil1*, and *piwil2*, were observed in the fish of different genotypes, except *cyp17a1-/-;ar-/-;npgr-/-* fish (**Figure 6C**). These results might indicate that the upregulated expression levels of germ cell markers are not critical for the restoration of the defective phenotypes of the testis organization and spermatogenesis caused by the lack of androgen signaling in these various mutant lines (**Tang et al., 2018**). The Insl3 receptors, *rxfp2a* and *rxfp2b*, expressed in type A spermatogonia and Sertoli cells/myoid cells, respectively, were also decreased in the *cyp17a1-/-;ar-/-;npgr-/-* fish compared with that in the control males (**Figure 6B and C**), suggesting compromised Insl3 signaling in *cyp17a1-/-;ar-/-;npgr-/-* fish. The upregulated expression of the retinoic acid-degrading enzyme, *cyp26a1*, decreased GSI and disrupted testis morphology in the *cyp17a1-/-;ar-/-;npgr-/-* fish (**Figures 6D and 4H** and I), correlating with the observations in the *insl3-/-* males (**Crespo et al., 2021**).

The sperm fertilities of the *ar+/+* males and *ar-/-* males after treatment with DHP were also measured. Unfortunately, both experiments showed impaired fertility after DHP treatment via artificial fecundation assays with WT females (**Figure 3—figure supplement 1**). Based on our observations and several previous publications, the chemical reagents of sex steroids used in the animal administration would result in developmental abnormalities, such as the E2 usually leads to abnormal body growth, enlarged abdomen and liver, and a large amount of liquid in the peritoneal cavity induced by 10 nM E2 treatment from 20 dpf to 40 dpf (**Chen et al., 2017**), and 10 μg/L (36.71 nM) E2 treatment from 18 dpf to 90 dpf, as well as DHP treatment in the present study (data not shown). On the other hand, based on the results in **Figure 2B**, the dynamic levels of the elevated DHP seen in the *cyp17a1-/-* fish might suggest the failure of rescued fertility due to being unable to precisely match in vivo DHP dynamic levels or the potential side effects of the constant levels of exogenous DHP exposure. However, partial restoration of the testicular phenotypes of *ar-/-* males via DHP administration supports the mechanism by which an enhancement of progestin signaling pathway is an alternative to testis organization and spermatogenesis to androgen signaling (probably via Insl3) in *cyp17a1-/-* fish. Therefore, a high level of endogenous or exogenous DHP can compensate for androgen signaling insufficiency to facilitate testis organization and spermatogenesis (**Figure 7C–E**).

In summary, our present study with a series of sex steroid signaling deficiency models and DHP administration provides compelling evidence demonstrating that an augmentation of progestin/nPgr signaling can sufficiently compensate for proper spermatogenesis and testis organization when androgen signaling is depleted in zebrafish.

## Materials and methods

**Key resources table**

| Reagent type (species) or resource | Designation | Source or reference | Identifiers | Additional information |
|---|---|---|---|---|
| Genetic reagent (*Danio rerio*) | *cyp17a1* knockout | PMID:30202919 | RRID:ZFIN_ZDB-GENO-191029-6 | Dr. Zhan Yin (Institute of Hydrobiology, Chinese Academy of Sciences) |

*Continued on next page*

Continued

| Reagent type (species) or resource | Designation | Source or reference | Identifiers | Additional information |
|---|---|---|---|---|
| Genetic reagent (*D. rerio*) | *fshβ* knockout | PMID:30202919 | RRID:ZFIN_ZDB-GENO-191029-7 | Dr. Zhan Yin (Institute of Hydrobiology, Chinese Academy of Sciences) |
| Genetic reagent (*D. rerio*) | *ar* knockout | PMID:29849943 | RRID:ZFIN_ZDB-GENO-190307-7 | Dr. Wuhan Xiao (Institute of Hydrobiology, Chinese Academy of Sciences) |
| Genetic reagent (*D. rerio*) | *npgr* knockout | PMID:27333837 | RRID:ZFIN_ZDB-GENO-170907-1 | Dr. Xiaochun Liu (Sun Yat-Sen University) |
| Chemical compound, drug | 11-KT | Efebio | Cat# E092432 | |
| Chemical compound, drug | DHP | TRC | Cat# P712080 | |
| Chemical compound, drug | P4 | Aladdin | Cat# P106426 | |
| Chemical compound, drug | DMSO | Sigma-Aldrich | Cat# D2650 | |
| Chemical compound, drug | TRIzol reagent | Ambion | Cat# 15596 | |
| Commercial assay or kit | T ELISA kit | Cayman Chemicals | Cat# 582701 | |
| Commercial assay or kit | 11-KT ELISA kit | Cayman Chemicals | Cat# 582751 | |
| Commercial assay or kit | First-strand cDNA synthesis kit | Fermentas | Cat# K1621 | |
| Antibody | Anti-Vasa (rabbit polyclonal) | GeneTex | Cat# GTX128306 | RRID:AB_2847856 IF: (1:500) |

## Animals

The zebrafish were maintained under standard conditions at 28.5°C. *cyp17a1*, *ar*, *fshβ*, and *npgr* knockout lines were established as previously described (*Tang et al., 2016*; *Yu et al., 2018*; *Zhai et al., 2018*; *Zhai et al., 2017*). The *cyp17a1* heterozygote was crossed with an *ar* heterozygote to generate the *cyp17a1/ar* double heterozygous fish. The *cyp17a1/ar* double heterozygote was crossed with the *fshβ* heterozygote to generate the *cyp17a1/ar/fshβ* triple heterozygous fish. The *cyp17a1/ar* double heterozygote was crossed with the *npgr* heterozygote to generate the *cyp17a1/ar/npgr* triple heterozygous fish. Double and triple heterozygous fish were used to generate homozygous fish. Male zebrafish were sampled and analyzed from the genotypes mentioned above at 6 mpf. Fish genotypes from the population were examined as previously described (*Tang et al., 2016*; *Yu et al., 2018*; *Zhai et al., 2018*).

## Steroid hormones measurement using UPLC-MS/MS

The concentrations of 11-KT, DHP, and P4 from whole-body lysates were measured using UPLC-MS/MS (*Wang et al., 2022*). Briefly, the whole body of each fish was placed in a tube containing magnetic beads in a high-speed vortex destroyer instrument (Tissue Cell-destroyer 1000, Xinzhongke viral Disease Control Bio-Tech Ltd, Hubei, China), and then ice-cold acetonitrile (1.5 mL) was added. The centrifugation was performed at 4°C with the highest speed after the high-speed vortex and ultrasonic disruption. The upper layer was placed into a glass tube and evaporated at 30°C under a gentle stream of nitrogen. After dissolving with 0.6 mL of methanol, 2.4 mL of double-distilled water was added, mixed, and purified with C18 solid-phase extraction cartridges (100 mg sorbent per cartridge, RNSC1003-C18, Lvmeng, Jiangsu, China) for further measurement using UPLC-MS/MS (ACQUITY UPLC, Quattro Premier XE, Waters, USA). Then the purified products were evaporated with nitrogen and dissolved with 40% methanol. The 11-KT (E092432, Efebio, Shanghai, China), DHP (P712080, TRC, North York, Canada), and P4 (P106426, Aladdin, Shanghai, China) at a series of concentrations were used as standard samples for standard curve establishment. The standard samples (powders) were dissolved in DMSO (D2650, Sigma-Aldrich, St. Louis, MO) to a concentration of 1 mg/mL, and the gradient dilution with 40% methanol was performed. $R > 0.995$ was considered as qualified linear of the gradient dilution of the standard samples.

## Steroid hormones measurement using ELISA

The concentrations of T and 11-KT in testis were measured using commercial ELISA kits (T: 582701; and 11-KT: 582751, Cayman Chemicals, Ann Arbor, MI). Briefly, testis samples were isolated and

homogenized in phosphate-buffered saline (PBS). After homogenization, an organic solvent was used to extract the sex steroids according to the manufacturer's instructions. The layers were separated by vortexing and centrifugation, the organic layer was transferred to a fresh tube, and the extraction was repeated four times. The organic part was evaporated by heating to 30°C under a gentle stream of nitrogen. Finally, the extracts were dissolved in 200 µL ELISA buffer and prepared for measurement according to the manufacturer's instructions.

## Histological analyses (H&E staining) and spermatozoa number analysis

The dissected testes were fixed in 4% paraformaldehyde in PBS at room temperature, followed by dehydration and infiltration. The sectioning and staining procedures were performed as previously described (Lau et al., 2016). Briefly, the samples were embedded and processed for transverse sectioning in paraffin using a Leica RM2235 microtome (Leica Biosystems). Paraffin sections (5 µm in thickness) were mounted on slides, deparaffinized, rehydrated, and washed with deionized water. The sections were stained with hematoxylin and eosin (H&E), dehydrated, mounted, and visualized under a Nikon Eclipse Ni-U microscope (Nikon, Tokyo, Japan). Scale bars are provided for each image. For the spermatozoa number analysis, ImageJ software was used. Briefly, the area in each integrative lumen of seminiferous tubules containing spermatozoa was cut and saved as a new image file for further analysis. After the image was reloaded, the image was transferred to 8-bit gray, and the threshold was selected for sperm selection. The particle number was analyzed after the defining point under the binary submenu to dissect the stacked sperm.

## DHP administration

The population of ar heterozygotes containing ar+/+, ar+/-, and ar-/- fish was administered with DHP (P712080, TRC) from 50 to 90 dpf. Briefly, the population from ar heterozygotes was placed in a 3.5 L tank containing DHP (0.067 µg/mL). After the treatment, each fish was genotyped with a tail fin cut and subjected to anatomical examination, GSI analysis, testis HE staining, and spermatozoa number analysis.

## Immunofluorescence staining

Immunofluorescence staining was performed using an anti-Vasa rabbit polyclonal antibody (GTX128306-S, GeneTex, Irvine, CA) as the primary antibody (Zhu et al., 2019). Fluorescein (FITC)-conjugated goat anti-rabbit IgG (H+L) was used as the secondary antibody (SA00003, Proteintech, Rosemont, IL). As previously described, zebrafish testes were fixed, embedded, sectioned, and stained using standard protocols (Zhu et al., 2019). Nuclear DNA was stained with 4',6-diamidino-2-phenylindole

**Table 1.** Primers for qPCR used in this study.

| Gene | Primer direction and sequence (5'–3') | Reference |
|---|---|---|
| | F: GATCCGACTGCTGGATAGAAACA | |
| hsd3β1 | R: CCCGGCAATCATCAAGAGA | Crespo et al., 2021 |
| | F: CGGACGGTGGTCGCATCGTG | |
| insl3 | R: CTCTCTGGTGCACAACGAG | Zhai et al., 2018 |
| | F: CCAGGATTCATGCTGAAGGTG | |
| igf3 | R: CTACGAGCTGCTCCAGGTTTG | Zhai et al., 2018 |
| | F: TCGTGGAAGCTTTTCGGAACCGG | |
| dnd | R: TGTCCTCGACGCGCTTGGAC | Lin et al., 2017 |
| | F: TGGGCTTGCCGTTCATTG | |
| cyp26a1 | R: CATGCGCAGAAACTTCCTTCTC | Crespo et al., 2021 |
| | F: GCCGTCCCACCGACAAG | |
| ef1a | R: CCACACGACCCACAGGTACAG | Crespo et al., 2021 |

F = forward; R = reverse.

(DAPI). Sections were visualized using ×40 objective lenses of an NOL-LSM 710 microscope (Carl Zeiss, Germany). Scale bars are provided for each image.

## Transcriptome analyses

Testis RNA was isolated from the testes of zebrafish by extraction with TRIzol reagent (15596, Ambion, Austin, TX). Using an Illumina NovaSeq 6000 system, RNA-seq reads were generated by sequencing. High-quality mRNA reads were mapped to the *Danio rerio* genome (GRCz11) using HISAT2 (version 2.2.4, http://daehwankimlab.github.io/hisat2/). Differential expression analysis was performed using the DESeq2 package (v1.30.1) with a fold change of 2 and a p-value cutoff of 0.05. Venn diagram analysis of differentially expressed genes was performed by the R package VennDiagram (version 1.7.1). A heatmap for candidate genes was plotted in R (version 4.1.0) using the heatmap package.

## Quantitative real-time PCR (qPCR)

Independent of the RNA samples for transcriptome analyses, another group of RNA samples was extracted and used for cDNA synthesis for qPCR to confirm the transcriptome results. According to the manufacturer's instructions, a total of 1.5 µg of RNA template was used for reverse transcription to synthesize cDNA using a first-strand cDNA synthesis kit (K1621, Fermentas, Waltham, MA). The qPCR primers for *hsd3β1*, *insl3*, *igf3*, *dnd*, *cyp26a1*, and *ef1a* were used as previously described and are listed in *Table 1*. For amplification, the TransStart Tip Green qPCR SuperMix (AQ141-01, TransGen, Beijing, China) and Bio-Rad real-time system (Bio-Rad Systems, Berkeley, CA) were used. All mRNA levels were calculated as the fold expression relative to the housekeeping gene *ef1a* and expressed as a fold change compared to the control group.

## Statistical analysis

All the experiments were conducted at least two times. Data were analyzed using GraphPad Prism 8 software. All results are reported as mean ± SD. The statistical significance of differences was determined using Student's *t*-test for paired comparisons and one-way ANOVA, followed by Fisher's LSD test for multiple comparisons. For all statistical analyses, $p < 0.05$ indicated a significant difference. Significant differences marked with asterisks and letters were analyzed using Student's *t*-test for paired comparisons and one-way ANOVA, followed by Fisher's LSD test for multiple comparisons, respectively.

## Acknowledgements

We thank Mr. Shibo Ma, Institute of Hydrobiology, Chinese Academy of Sciences, for taking care of the zebrafish stocks. We thank Jun Men (Center for Instrumental Analysis and Metrology, Institute of Hydrobiology, Chinese Academy of Sciences) for technical assistance with UPLC-MS/MS. This work was supported by the National Key Research and Development Program, China (no. 2018YFD0900205), the Pilot Program A Project from the Chinese Academy of Sciences (no. XDA24010206) and National Natural Science Foundation, China (no. 31972779, no. 31530077, and 31702027), the Youth Innovation Promotion Association of CAS (2020336), and the State Key Laboratory of Freshwater Ecology and Biotechnology (2016FBZ05). The funders had no role in study design, data collection and interpretation, or the decision to submit the work for publication.

## Additional information

### Funding

| Funder | Grant reference number | Author |
| --- | --- | --- |
| National Key Research and Development Program of China | 2018YFD0900205 | Zhan Yin |
| Chinese Academy of Sciences | Pilot Program A Project XDA24010206 | Zhan Yin |

| Funder | Grant reference number | Author |
| --- | --- | --- |
| National Natural Science Foundation of China | 31972779 | Gang Zhai |
| National Natural Science Foundation of China | 31530077 | Zhan Yin |
| National Natural Science Foundation of China | 31702027 | Xiangyan Dai |
| Youth Innovation Promotion Association of CAS | 2020336 | Gang Zhai |
| State Key Laboratory of Freshwater Ecology and Biotechnology | 2016FBZ05 | Zhan Yin |
| Research and Development | | Zhan Yin |
| Chinese Academy of Sciences | | Zhan Yin |
| Youth Innovation Promotion Association | 2020336 | Gang Zhai |

The funders had no role in study design, data collection and interpretation, or the decision to submit the work for publication.

## Author contributions

Gang Zhai, Data curation, Funding acquisition, Investigation, Methodology, Validation, Visualization, Writing – original draft; Tingting Shu, Investigation, Methodology, Validation, Visualization, Writing – review and editing; Guangqing Yu, Methodology, Resources; Haipei Tang, Wuhan Xiao, Xiaochun Liu, Resources; Chuang Shi, Investigation, Methodology, Visualization; Jingyi Jia, Investigation, Methodology, Software; Qiyong Lou, Investigation; Xiangyan Dai, Validation, Visualization; Xia Jin, Resources, Supervision; Jiangyan He, Methodology, Project administration, Resources; Zhan Yin, Conceptualization, Funding acquisition, Methodology, Project administration, Resources, Supervision, Writing – review and editing

## Author ORCIDs

Tingting Shu http://orcid.org/0000-0002-3020-9329
Wuhan Xiao http://orcid.org/0000-0002-2978-0616
Zhan Yin http://orcid.org/0000-0002-7969-3967

## Ethics

Animal experimentation: All fish experiments were conducted in accordance with the Guiding Principles for the Care and Use of Laboratory Animals and were approved by the Institute of Hydrobiology, Chinese Academy of Sciences (Approval ID: IHB 2013724).

## Decision letter and Author response

Decision letter https://doi.org/10.7554/eLife.66118.sa1
Author response https://doi.org/10.7554/eLife.66118.sa2

# Additional files

## Supplementary files

• Supplementary file 1. The gene list and basemean of the total of 148 differentially expressed genes in the overlapped region of the Venn diagram. (a) The gene list and basemean of the total of 148 differentially expressed genes in the overlapped region of the Venn diagram. (b) The gene list and basemean of the total of 111 differentially expressed genes in the overlapped region of the Venn diagram.

• Transparent reporting form

## Data availability

The knockout fish and genes involved in this study have been cited and clearly listed in the references. The transcriptomics raw data files in this article are available in Sequence Read Archive (SRA) at NCBI, and the BioProject is PRJNA796639.

The following dataset was generated:

| Author(s) | Year | Dataset title | Dataset URL | Database and Identifier |
| --- | --- | --- | --- | --- |
| Chinese Academy of Sciences | 2022 | Augmentation of progestin signaling rescues testis organization and spermatogenesis in zebrafish with the depletion of androgen signaling | https://www.ncbi.nlm.nih.gov/sra/?term=PRJNA796639 | NCBI Sequence Read Archive, PRJNA796639 |

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
