## [Editor Report]

Deletion of the androgen receptor leads to testicular malformation and defective spermatogenesis in zebrafish. Paradoxically, simultaneous deletion of the androgen receptor and cyp17a1, a key enzyme in the androgen synthesis pathway, leads to testicular hypertrophy and enhanced spermatogenesis. By analyzing a number of mutant zebrafish lines, the authors have provided new evidence suggesting that an elevation of progestin signaling can compensate for the loss of androgen receptor and progestin promotes testis organization and spermatogenesis via the nuclear progestin receptor.

---

## [Decision Letter]

**Decision letter after peer review:**

Thank you for submitting your article "Progestin signaling facilitates testicular development and spermatogenesis independently from androgen signaling in fish" for consideration by *eLife*. Your article has been reviewed by 3 peer reviewers, one of whom is a member of our Board of Reviewing Editors, and the evaluation has been overseen by Didier Stainier as the Senior Editor. The reviewers have opted to remain anonymous.

Summary

Previous studies from this group showed that androgen deficient (cyp17a1-/-) fish are all male with intact spermatogenesis. This is in good agreement with earlier findings by others: in fish, spermatogenesis can be supported by other than androgen-driven signaling pathways. While progestin-mediated stimulation of spermatogenesis in fish has been described previously including zebrafish, convincing genetic data and the underlying mechanisms are still lacking. In this study, using a number of zebrafish mutant fish lines, the authors provide genetic data suggesting a possible role of progestins (P and DHP) in regulating spermatogenesis via the nPgr. The reviewers feel that there are merits and potentials in this study. At present, however, this manuscript is still premature, descriptive, and lacks mechanistic insights. Several major issues/concerns were raised by the reviewers.

The following concerns and suggestions should be addressed.

Essential revisions:

1) The lack of anatomical and histological data on the cyp17a1/npr double mutant fish. In contrast to other mutant lines, only the expression of selected testicular genes was studied in the cyp17a1/npr double mutant fish. In order to answer the question, the anatomical and histological data of these testis must be included.

2) In zebrafish, both Fsh and Lh can signal through the Fsh receptor. Both Fsh and Lh levels are elevated in the crp17a1 mutant fish. Therefore, the crp17a1/ar/fsh triple mutant data is insufficient. The Fsh receptor instead of Fsh β subunit should have been removed in the triple mutant analysis.

3) A key issue raised is the reported P and DHP levels and the steroid assays used. One reviewer felt that the reported progestin concentration may be too low to activate the Ar. Another suggested to measure P and DHP levels using LC/MSMS, which is preferable when measuring P and DHP in this low concentration range as in zebrafish. Whole body extracts contain several compounds co-extracted with steroids that can disturb the assays. The extraction process will lose some P and DHP. If the authors choose to use the same kit, the approach used to extract steroids from tissue samples needs to be specified and quality control experiments/data should be included.

4) The authors showed that P4 and DHP treatment increased GSI and spermatozoa number. It is pointed out that the doses used in P and DHP treatments may be too low to be effective or the P and DHP quantification may be way off. It would also be important to assess the P4 and DHP concentrations in pharmacologically treated fish. Did they look at the mature sperm number, motility and fertility?

5) Relevant in normal physiology. It would have been useful to compare the presented directly with npgr-/- mutants throughout and compare the cyp17a1-/-; npgr-/- to the ar-/-; npgr-/- and the single mutants. This plus treatment would have clearly answered if progestins play a role in normal physiology. Furthermore, is it possible to inhibit progestin synthesis/secretion in cyp17a1-/- fish or use progestin antagonists (e.g., mifepristone, ORG3170 etc.?) If possible, it will provide further evidence that the elevated levels of progestins is indeed critical.

6) Histological analysis: Figures 2, 3 and 4 show quantitative data (the ordinates are labelled with, "number of sperm per cyst"). However, it is not clear how the quantitative histological data were obtained and what exactly they show.

7) Immunofluorescence staining: The authors used an antibody against mouse vasa protein on zebrafish testis sections. The antibody used has not been characterized. The ICC procedure has not been described. No proof is provided that the antibody against a mammalian protein reliably detects zebrafish vasa protein.

8) RNA extraction and qPCR: The authors report to have used a single 'housekeeping gene' (ef1a). However, the cellular composition of testis tissue varies in the different mutants, so that it cannot be excluded that also the readings for ef1a changed, depending on the genotype.

9) Statistical analysis: The statistical analyses appear incomplete and may even be flawed. This makes it difficult to assess reproducibility of the quantitative measurements, the claims and conclusions made in this study. The authors should consult a statistician to resolve this issue.

10) Titles: It is surprising that the authors repeatedly mentioned "testicular development", including in the title, since the data presented covers adult fish only.

11) A number of substantive concerns were raised with the statements/overstatements in the text. These should be addressed.*Reviewer #1:*

The authors were intrigued by the observation that loss of the androgen receptor (ar) in zebrafish has a clear impact on spermatogenesis while spermatogenesis remained intact after blocking androgen production through loss of the enzyme cyp17a1. This supports an earlier conclusion also proposed by others: in fish, spermatogenesis can be supported by other than androgen-driven signaling pathways. Since the authors found progestin levels to be higher in cyp17a1 mutants, and since results published previously by others showed that progestins stimulated different aspects of spermatogenesis (spermatogonial development, meiosis, sperm capacitation) in different fish species (eel, tilapia, trout, zebrafish), the authors asked if removing also the nuclear progesterone receptor (npr) from cyp17a1 mutants would result in a spermatogenesis phenotype in zebrafish after all. To answer this valid question, the authors generated mutant lines to seek support for the hypothesis that progestin signaling via its nuclear receptor can maintain spermatogenesis in androgen-depleted cyp17a1 mutants. Also ar-KO zebrafish were subjected to progesterone treatment, although the reasons for that were less clear. Surprisingly, the mutant most pertinent to the main question, the cyp17a1/npr double mutant, was only used to analyze the expression of selected testicular genes. No anatomical or histological data were included, in contrast to all other lines, thus leaving the main question unanswered regarding effects on spermatogenesis. Instead, triple mutants were generated by removing also the androgen receptor (cyp17a1/npr/ar) or the Fsh β subunit (cyp17a1/npr/fshb). Unfortunately, the loss of a third gene does not allow to draw "clean" conclusions regarding the consequences of the loss of npr in males unable to produce androgens; also, it is not clear what led to the choice of ar or fshb as third genes to be removed. These points are further discussed in the following.

Follicle-stimulating hormone β subunit (fshb)

Referring to works published by others, the authors state correctly that Fsh can stimulate spermatogenesis independent of sex steroid signaling. Since elevated levels of both Lh and Fsh have been reported in cyp17a1 KO zebrafish, the authors decided to remove the Fsh β subunit gene (reminiscent of work in their 2018 paper), attempting to study the contribution, if any, of Fsh signaling to maintaining spermatogenesis. However, in the cyp17a1/ar/fshb triple mutant, the Fsh receptor is still present. Since the also elevated Lh levels can cross-activate the Fsh receptor (Xie et al., 2017, DOI: 10.1530/JOE-17-0079), it is conceivable that Fsh receptor-mediated signaling remained relevant. Therefore, it seems that removal of the receptor, instead of the ligand, would have been the option to choose for investigating the role of Fsh signaling.

The authors concluded that removing fshb had no effect on spermatogenesis, referring to GSI levels being similar to controls (L290). However, the authors did not point out that in the cyp17a1/ar/fshb triple mutant, GSI values were halved compared to the cyp17a1/ar double mutant, i.e. removal of fshb apparently did have an effect.

Androgen receptor

The authors mention the possibility that elevated levels of low affinity ligands might activate the Ar in the absence of androgens, such as the elevated progestin levels of found in cyp17a1 mutants. Based on the data presented in Figure 1, the reviewer calculated the combined concentrations of P and DHP to be ~60 nM (data from Figure 1). This is 10- to 20-fold lower than the progestin concentration required to induce some activity of the zebrafish Ar; moreover, the progestin-induced Ar activity is only ~1/10th of the activity induced by androgens (based on the data cited by the authors as de Waal et al., 2008). Hence, available information suggested that progestin concentrations were too low and could only marginally activate the Ar, and therefore seem irrelevant for cross-activating the zebrafish Ar. The reviewer wonders about potential other reasons for carrying out the experiments shown in Figure 2.

A question requiring attention is the treatment with progestins (Figure 2) also relating to Figure 1. It shows the endogenous progestin levels in ar-KO mutants being ~5-10ng/g = 5-10µg/kg, equivalent to 5-10µg/L. In Figure 2 and the associated text, the authors report clear effects following exposure of ar-KO fish to P or DHP at a concentration of 0.1µg/L. Reporting clear treatment effects using progestin concentrations 50-100 times lower than the endogenous concentrations would require a careful discussion.

Other results obtained with fish carrying a mutated ar await being discussed. For example, the authors find, similar to others, that spermatogenesis is compromised in ar-KO zebrafish and claim (L193 and following) that testicular gene expression analysis (Figure 6) supports this finding. However, the five 5 germ cell genes quantified seem clearly elevated in ar-KO fish showing compromised spermatogenesis after loss of the ar. How to reconcile increased germ cell gene expression with compromised spermatogenesis remains unclear. Similarly, germ cell gene expression is reported to be upregulated in the cyp17a1/npr/ar triple mutants compared to cyp17a1/npr double mutant, i.e. the combination of impaired spermatogenesis and increased germ cell gene expression following removal of the ar gene comes up again. This time, however, not all five germ cell genes but only four were responding, and it is unclear why is there an "odd one out" in this specific genetic model.

More comprehensive discussion is perhaps also required regarding the Tgf β family member Amh. The authors did discuss the role of Amh in spermatogonial differentiation, but may have overlooked Amh-promoted transition into meiosis (https://doi.org/10.1016/j.mce.2020.110963).

The authors repeatedly make the point, including in the title, that progestin signaling stimulates spermatogenesis independently from androgen signaling. This wording is not sufficiently precise. The genetic evidence presented is based on fish that are unable to produce androgens. This does not exclude the possibility that the Ar protein can exert biological activity in the absence of androgens (or any other type of ligand). Ligand-independent Ar action may indeed be related to changes in gene expression induced by removing the ar from the cyp17a1/npr double mutants (see above). Therefore, the conclusion/point (progestin stimulates spermatogenesis independently from androgen signaling) should be restricted to fish unable to produce androgens. Another reason to do so is that in the presence of androgens (the physiological situation), androgen/Ar-mediated effects may very well interact with progestin/Npr effects on spermatogenesis.

A repeatedly occurring statement requiring attention is the mentioning of testis development or testicular development in the manuscript. It would be appropriate to remove all of this from the manuscript, including the title, because respective data is not presented. The datasets in the manuscript refer exclusively to adult males.

Another point refers to an alternative understanding of the phenotype of the cyp17a1 mutant. First mentioned in L96-97, this mutant is described as presenting stimulated or enhanced spermatogenesis. The reviewer assumes that the authors refer to the GSI levels and spermatozoa number being higher than in WT adults older than 3 months. The age is relevant since in young adults up to 3 months of age, the GSI is still similar to WT controls. However, neither in the previous study (Zhai et al., 2018) nor in the present manuscript, spermatogenic activity has been studied in detail. The fact that the phenotype of increased GSI/sperm number developed with time after 3 months of age (as described in Zhai et al., 2018) and the histological pictures shown in Zhai et al., 2018 and in the present manuscript, suggests that the weight gain is based on progressively accumulating spermatozoa. Since cyp17a1 mutants don't show reproductive behavior/spawning, the reviewer understands the slowly increasing GSI by progressively accumulating sperm formed at a normal rate. Claiming enhanced spermatogenesis requires research into the dynamics of this cellular development that is missing as yet.

The accumulation of sperm may also explain increased DHP levels. The enzyme converting 17aP to DHP, 20bHSD, is highly active in spermatozoa of salmonid and cyprinid species, including the zebrafish relatives carp and goldfish. In this regard, elevated DHP levels may reflect an "accidental side-effect" of sperm with their associated 20bHSD activity, accumulating in the testes of cyp17a1 mutants, and then metabolizing the substrate 17aP that is available in high levels due to the loss of cyp17a1.

Overall, the reviewer considers it as a pity that the manuscript does not report also first results regarding mechanisms underlying the progestin-stimulated maintenance of spermatogenesis in cyp17a1 KO zebrafish. Such data would elevate the manuscript from its descriptive nature. Nevertheless, the authors have generated valuable genetic models that may allow, in the future, not only to provide the previously reported, progestin-mediated stimulation of spermatogenesis with a firm genetic basis in zebrafish, but also to broaden our knowledge on the mechanisms underlying progestin effects on spermatogenesis.

Materials and methods section

Steroid assays: The approach used to extract steroids from tissue samples needs to be specified and quality control experiments/data are missing. Contrary to the authors' statement, the procedure is not specified by the manufacturer. Looking up the manufacturer's specifications showed that these assays are meant for aqueous samples (plasma, serum, whole blood, urine samples, in vitro medium samples are mentioned, but not tissue or whole body extracts). Therefore, validation of the techniques for a new type of samples is required. This also includes the point that the authors apparently did not control for procedural losses. In addition, whole body extracts contain several compounds co-extracted with steroids that can disturb the assays, as indicated by the manufacturer, who suggested to remove the impurities even from much less complex samples such as blood plasma extracts, before introducing the samples into the assay. Has this been done? Finally, the authors do not report validation and standardization of the assays (e.g. extraction efficiency and potential correction for procedural losses, or reliability [i.e. is a spiked amount of steroid found back at the expected concentration?]). As reported now, results on steroid quantifications cannot be accepted as sufficiently reliable or accurate.

Histological analysis: Figures 2, 3 and 4 show quantitative data (the ordinates are labelled with, "number of sperm per cyst"). However, the respective M and M section does not detail how the data was obtained and processed. Regarding the label on the ordinate, please refer to literature on spermatogenesis in fish, the point being that after completion of spermiogenesis, the spermatogenic cysts open and germ cells are released by the Sertoli cells into the lumen of the spermatogenic tubules. From then on, the germ cells are called spermatozoa, which are no longer in cysts. Hence, "sperm per cyst" is a contradiction in terms, leading to the question if "sperm" or "per cyst" is not correct? These points require clarification for the data to be kept in the manuscript. Also the legend to Figure 5 requires attention in this regard. The authors state that scale bars are provided in each image – this is not correct, since for example on Figure 2, there are no scales; the scales on Figure 4, on the other hand, are probably not correct. In L342, please replace size by thickness.

P4 and DHP treatment: Please refer to the point made above in this context in Public Review (100-fold lower concentration used for treatments than found in the fish).

Immunoflourescence staining: The authors report to have used an antibody against mouse vasa protein on zebrafish testis sections. The antibody used has not been characterized. The ICC procedure has not been described. No proof is provided that the antibody against a mammalian protein reliably detects zebrafish vasa protein. In the Results section, the authors state in L182 that the ICC data demonstrate "…a reduced number of spermatozoa…in ar-/- males…but increased number of spermatozoa … in … cyp17a1 fish". However, spermatozoa do not express vasa protein. Moreover, since the ICC results in Figure 5 have neither been quantified nor normalized, statements referring to differences in cell numbers should be avoided.

RNA extraction and qPCR: The authors report to have used a single 'housekeeping gene' (ef1a). However, the cellular composition of testis tissue varies in the different mutants, so that it cannot be excluded that also the readings for ef1a changed, depending on the genotype. The request is to include the ef1a data as a separate graph in Figure 6 (e.g. as Figure 6K) for the different genotypes and subject them to statistical analysis. If differences are detected between genotypes, it is possible to use the approach explained in the following. For the future (or in case ef1a shows changes here), consider using at least three different reference genes, showing a large range of expression levels, and calculate qPCR results using their geometric mean as normalization.

Statistical analysis: This is one of the more critical points in the manuscript, since the potential problems here prevent evaluation of all quantitative data presented in the manuscript. The number of replicates per experiment/treatment group has not been provided. It is mentioned that experiments were repeated three times. However, how were the data obtained from these three experiments then used for statistical analysis and plotting of the graphs? In most graphs presenting quantitative data, the authors compare more than 2 groups with each other, usually 4-6 groups. The authors state in L369 that "differences were assessed using Student's t-test." With this test, however, only differences between two groups can be assessed, under the condition that the data is distributed normally. However, neither was data distribution tested, nor were there only two groups. Therefore, all statistical analyses need to be repeated.

Introduction

As indicated above, this following list mentions selected points independent of the re-analysis of the data. The term "selected" is used also because the number of points requiring comments is high and the list is far from complete. Hence, the authors will have to identify the points not mentioned below during a thorough re-writing process.

In the reviewer's opinion, the complete 1st paragraph of the Introduction can be removed.

L49-56: This sentence contains several unclear or awkward statements (e.g. "the reproduction research" – grammar; "advantage of gonadal dissection" – ??; "successfully adopted KO strategy" – applies to virtually thousands of species; "pharmacological utilization" – ???).

Conclusions in L57-61: the 1st conclusion is irrelevant in the present context; the 2nd conclusion is in part wrong (see following point); the 3rd conclusion is irrelevant in the present context

L58-61: Contrary to the authors' statement androgen signaling is not essential for testicular differentiation and development or for spermatogenesis in fish and the papers cited do not make this claim either. After all, the authors did study testis tissue from ar-KO mutants and this tissue contained all types of germ cells including sperm (e.g. Figure 2K). This wrong statement is repeated in L62 (wrong regarding testis development in fish).

L79-82: A series of statements is made but not supported by citations.

L103-104: The authors state that "administration of P4 and DHP effectively restored … GSI and spermatozoa number…". However looking at the graphs in Figure 2 G and N, the formulation is not correct since the controls showed higher levels (pending the re-analysis of the data, significantly higher in the controls). This type of error/overstatement occurs frequently.

L110: Another example for an overstatement is to use the term "arrested" to describe the effect of ar loss on spermatogenesis. While this is correct for mammals, it is not for zebrafish (see the authors' data in Figure 2K, showing an ar-KO testis with all types of germ cells including spermatozoa, i.e. no arrest).

Overall, the introduction contains passages that rather read like Results or Discussion section. Please amend.

Results and Discussion

Both sections contain statements that would require comments. However, the contents of these statements have been touched upon already, in one way or another, in the points discussed above. Therefore, the reviewer does not make specific comments to these sections, also considering the statistical/data uncertainties and technical questions remaining to be solved, which will likely result in several changes in these two sections.

*Reviewer #2:*

The authors present a paper suggesting that progestin facilitates testicular development in zebrafish. Overall, the paper tries to explain interesting observations made in their cyp17a1-/- lines that are different compared to fish with androgen deficiency and androgen resistance. Interestingly, the specific block in steroidogenesis led to an increase of progestins and the authors work up the hypothesis that progestins represent an androgen-independent regulator of testicular development and spermatogenesis. Is this a compensated "disease" state or a mechanism that is relevant under normal physiological circumstances?

The authors present quite some detailed analysis of a structured combination of various mutants/ double/ triple mutants to explore their hypothesis, which appears a major strength of the paper. There some aspects such as the effect pf progestin treatment on wildtypes and the discussion if these are physiological, pathophysiological or pharmacological effects that would need further analysis and clarification. A major weakness of the paper is that no direct comparison with npgr-/- lines have been made to explore the questions not only from the angle of androgen deficiency and resistance with progestin excess.

Overall this paper might form the basis some interesting novel research investigating the interplay between androgens and progestins in zebrafish gonadal development and function. The work is convincing, but despite the fact that progestins are known to play an important role in spermatogenesis, it is remains unclear if progestins play the same significant roles in wildtype fish.

Despite a number of very interesting experiments that clearly show a role progestins in testicular development in androgen-deficient/ resistant zebrafish, it remains unclear if this is relevant in normal physiology or only to explain overserved phaenomena in mutant lines. It would have been useful to compare the presented directly with npgr-/- mutants throughout and compare the cyp17a1-/-; npgr-/- to the ar-/-; npgr-/- and the single mutants. This plus treatment would have clearly answered if progestins play a role in normal physiology. Importantly, how high are concentrations in the treated zebrafish? Are these physiological or pharmacological effects?

Overall, it appears that the progestin excess does not compensate it rather leads to hyperplastic testes, which in itself appears interesting and could be further addressed. Could this be relevant to answer environmental questions/ exogenous progestin exposure?

Lines 253-257: This raises the key question of progestins are really required for testicular developments and spermatogenesis.

I disagree with this statement. The authors demonstrate is that progestins can override defective androgen synthesis and the effects on testicular development. It would have been very interesting see if sperm from the various mutant lines can fertilise eggs. In addition, the assessment of mature sperm count and their motility would have been vital to understand the phenotype better.

If progestins are required in a way that the authors suggest then a direct comparison with the npgr-/- in their representations would be very useful I not essential.

It remained unclear why effects on the hypothalamic pituitary gonadal axis have not been assessed? This could provide further evidence about the degree of compensation by progestins. It would have been interesting if there could be synergistic effect of gonadotrophs or if there is a downregulation or even more interestingly if lh and fsh are differentially affected.

This could be very different to the upregulation of the HPG axis in recently described androgen deficiency fish is secondary. Parts of the discussion are not entirely clear as it cannot be expected to restore testicular development in other species. The message of this part of the discussion appears not entirely clear.

The steroid assays are not well described. How have they been validated? Assays to measure all compound by LC/MSMS are available and preferable when measuring analytes in this low concentration range as in zebrafish. The use of ELISAs for specific hormones appears outdated and less robust than LC/MSMS assay. This appears to be a real weakness of the paper.

It would be very important to assess the P4 and DHP concentrations in pharmacologically treated fish. The GSI does not seems to change; are the treated animals with larger testes heavier/ longer?

Over long stretches, I have asked myself why the authors did generate a triple mutant rather and only a double mutant cyp17a1-/-; ngpr -/-? This double mutant appears out of the blue in figure 6; why has no other data been provided on the double mutant? It appears vital to present data on that double mutant. As stated above several times it would have been important to get an idea about the phenotype of the npgr-/- to see what happens to spermatogenesis.

*Reviewer #3:*

Previous studies from this group showed that cyp17a1-/- fish are all male with enhanced testicular development and spermatogenesis. Since the testosterone (T) and 11-KT levels are lower in the mutant fish, it was postulated that there is an androgen-independent signaling pathway. In this study, the authors showed the levels of progestins (P4 and DHP) were elevated in cyp17a1-/- fish and treatment of ar-/- fish with exogenous P4 and DHP partially restored spermatogenesis and testicular development. By generating and analysis of cyp17a1, ar, npr double and triple mutant fish lines, they concluded that the elevated levels of progestins functions as an additional and Ar-independent pathway regulating zebrafish spermatogenesis and testicular development.

Overall, the findings are new and interesting. The genetic rescue crossing data are quite convincing. The manuscript is well written and the data nicely presented.

The following are my comments and questions:

1) The authors showed that P4 and DHP treatment increased GSI and spermatozoa number. Did they look at the mature sperm number, their motility and fertility in the treated fish? If yes, please include the data. If not, please discuss possible outcome.

2) Is it possible to inhibit progestin synthesis/secretion in cyp17a1-/- fish or use progestin antagonists (e.g. mifepristone, ORG3170 etc.? If possible, it will provide further evidence that the elevated levels of progestins is critical.

3) Figure 5 can be better described. For instance, how the authors identify SG, PSP, SSP, and SZ cells? This will help the general *eLife* readers. It will also be helpful to quantify the results.

4) The description and discussion about Figure 6 are not as clear as the other figures. For instance, the sycp3 results are different from those of vasa,, dnd, nano and piwill. What are these data telling us? Likewise, the patterns of star, hsd3b and cyp11a2 expression are different? What are these results saying and how do they related to the spermatogenesis phenotypes?

5) Line 154: "…confirm that the accumulated progestins, P4 and DHP, acts directly to facilitate testicular development and spermatogenesis …". Please remove the word "directly". This is too strong since no data suggesting the exogenously add P4 and DHP act indirectly or indirectly.

[Editors’ note: further revisions were suggested prior to acceptance, as described below.]

Thank you for resubmitting your work entitled "High level of progestin signaling facilitates testis organization and spermatogenesis independently from androgen signaling in fish" for further consideration by *eLife*. Your revised article has been evaluated by Didier Stainier (Senior Editor) and a Reviewing Editor.

The manuscript has been improved but there are some remaining issues that need to be addressed, as outlined below:

In this revised manuscript, the authors have provided a significant amount of new data, including

a) Generating the triple mutant fish and all single and double mutants and performing phenotypic and RNAseq analysis;

b) Re-measuring the steroid hormone levels using UPLC-MS/MS, and

c) Provided new data on Fshb levels. While they did not or could not generate Fsh receptor double and triple mutants, they made a good argument that should not affect their main conclusion. These new data have addressed most of the major concerns raised by the reviewers.

There, however, remaining issues that need to be addressed, as outlined below:

1) The current version of the manuscript is very difficult to read. Many sentences are hard to understand or can be interpreted in different ways. I suggest the authors to seek help from colleagues/professionals to edit the manuscript thoroughly. They can also choose to work with an *eLife* copyeditor to address this issue.

2) Another reason is presentational. Instead of presenting the data based on a chronological order, I suggest them to present the data following the order of Figure 7. For instance, it would be much easier for the readers to follow if they change Figure 3 into Figure 1, F1->2, and F2->.

3) The fertility data should be included.

4) The RNAseq data presentation. It is clear this is a huge amount of information. What is unclear is what do these data mean? I suggest the authors to dig into this dataset and present the data in relationship to the main conclusion/model (see Figure 7). For example, is it possible to see any changes in genes involved in the progestin/androgen biosynthesis (in addition to cyp17a1), nPgr target genes, and/or Ar target genes in these mutants?While it is okay to presenting all 8 groups together, it may be more meaningful to also compare the transcriptomic changes among different pairs of genotypes (in relationship to the main conclusion in Figure 7).

---

## [Author Response]

Essential revisions:1) The lack of anatomical and histological data on the cyp17a1/npr double mutant fish. In contrast to other mutant lines, only the expression of selected testicular genes was studied in the cyp17a1/npr double mutant fish. In order to answer the question, the anatomical and histological data of these testis must be included.

Good suggestion! According to this suggestion, we have analyzed the fish of the eight genotypes from the offspring of the crossed *cyp17a1*+/-;*ar*+/-;*npgr*+/- fish, including *cyp17a1*+/+;*ar*+/+;*npgr*+/+ males (control males), *ar*-/- males, *cyp17a1*-/- fish, *npgr*-/- males, *cyp17a1*-/-;*ar*-/- fish, *cyp17a1*-/-;*npgr*-/- fish, *ar*-/-;*npgr*-/- males, and *cyp17a1*-/-;*ar*-/-;*npgr*-/- fish (Line 183-186).

For the anatomical examinations and histological analyses, we found that GSI and spermatozoa number were both significantly decreased in the *cyp17a1*-/-;*npgr*-/- fish (Figure 4I and R). These results suggest that a potential compensatory role of progestin signaling exists in *cyp17a1*-/- fish. On the other hand, compared with the *cyp17a1*-/-;*ar*-/- fish, testis organization and spermatogenesis failed in the *cyp17a1*-/-;*ar*-/-;*npgr*-/- fish (Figure 4H and Q), as they displayed a similar pattern in testicular morphology and spermatogenesis as the *ar*-/- males and *ar*-/-;*npgr*-/- males (Figure 4B, G, K, and P). In addition, compared with the *cyp17a1*-/- fish and *cyp17a1*-/-;*ar*-/- fish, the observations of greater spermatogonia and spermatocytes, but fewer spermatozoa in the *ar*-/- males, *ar*-/-;*npgr*-/- males, and *cyp17a1*-/-;*ar*-/-;*npgr*-/- fish were observed in the histological analyses of testes (Figure 4J-Q). These results suggest that the compensatory pathway in promoting testis organization and spermatogenesis induced by *cyp17a1* knockout exists and is *npgr*-dependent (Line 183-198).

Transcriptome analyses of the testes of the control males, *ar*-/- males, *cyp17a1*-/- fish, *npgr*-/- males, *cyp17a1*-/-;*ar*-/- fish, *cyp17a1*-/-;*npgr*-/- fish, *ar*-/-;*npgr*-/- males, and *cyp17a1*-/-;*ar*-/-;*npgr*-/- fish, were performed (Line 215-232). For the markers of Leydig cells, the genes related to gonadal steroidogenesis, including *star*, *hsd3β1*, *cyp11a2*, and *cyp11c1*, were all increased in the testes of *ar*-/- males, *npgr*-/- males, *cyp17a1*-/-;*ar*-/- fish, *cyp17a1*-/-;*npgr*-/- fish, *ar*-/-;*npgr*-/- males, and *cyp17a1*-/-;*ar*-/-;*npgr*-/- fish. However, another gene of the Leydig cells, *insl3*, was significantly decreased in the *ar*-/- males, *ar*-/-;*npgr*-/- males, and *cyp17a1*-/-;*ar*-/-;*npgr*-/- fish (Figure 6A). For the markers of Sertoli cells, the genes including *sox9a*, *amh*, *Dmrt1*, *igf3*, and *rxfp2b* (the receptor of Insl3) were all decreased in the *ar*-/- males, *cyp17a1*-/- fish, *npgr*-/- males, *cyp17a1*-/-;*ar*-/- fish, *cyp17a1*-/-;*npgr*-/- fish, *ar*-/-;*npgr*-/- males, and *cyp17a1*-/-;*ar*-/-;*npgr*-/- fish (Figure 6B). For the markers of the germ cells, the genes, *dazl*, *vasa*, *dnd*, *nanos1*, *nanos2*, *piwil1*, and *piwil2*, were all upregulated in the *cyp17a1*-/-;*ar*-/-;*npgr*-/- fish, with different transcriptional expression patterns in the testes of other genotypes. Notably, *rxfp2a* (another receptor of Insl3) and the retinoic acid-degrading enzyme *cyp26a1* were downregulated and upregulated, respectively, in the *cyp17a1*-/-;*ar*-/-;*npgr*-/- fish (Figure 6C and D). The expression levels of *hsd3β1*, *insl3*, *igf3*, *dnd*, and *cyp26a1* were selected and further verified by qPCR (Figure 6E-I).

Then, the gene expression analyses of testis in fish with different genotypes were also discussed (Line 333-362). Increased expression of gonadal steroidogenesis-related genes, *star*, *hsd3β1*, *cyp11a2*, and *cyp11c1*, in Leydig cells was observed in the *ar*-/- males, *npgr*-/- males, *cyp17a1*-/-;*ar*-/- fish, *cyp17a1*-/-;*npgr*-/- fish, *ar*-/-;*npgr*-/- males, and *cyp17a1*-/-;*ar*-/-;*npgr*-/- fish. The upregulated expression of steroidogenesis-related genes in the *ar*-/- males was observed in the *ar* mutant males (Tang et al., 2018), which may be attributed to the positive feedback effect caused by androgen or progestin signaling insufficiency. Another marker of the Leydig cells, *insl3*, which has been reported to be essential for maintaining germ cell differentiation, was significantly decreased in the *ar*-/- males, *ar*-/-;*npgr*-/- males, and *cyp17a1*-/-;*ar*-/-;*npgr*-/- fish (which is the lowest), which are the only three genotype groups examined with impaired spermatogenesis and testis organization (Figure 6A). Considering that the downregulation of *insl3* expression has also been reported in both *cyp11c1* and *cyp11a2* mutant male zebrafish, which are associated with disorganized testicular structure and significantly decreased numbers of mature spermatozoa (Li et al., 2020; Zhang et al., 2020), it is reasonable to speculate that *insl3* may be a target that is synergistically regulated by androgen signaling and high level of progestin signaling. Among the upstream signals of the *insl3*, the role of the high level of progestin may be an alternative signaling pathway, other than androgen signaling, for the regulation of testis organization and spermatogenesis. The markers of Sertoli cells, *sox9a*, *amh*, *Dmrt1*, and *igf3* were decreased in the *ar*-/- males, *cyp17a1*-/- fish, *npgr*-/- males, *cyp17a1*-/-;*ar*-/- fish, *cyp17a1*-/-;*npgr*-/- fish, *ar*-/-;*npgr*-/- males, and *cyp17a1*-/-;*ar*-/-;*npgr*-/- fish (Figure 6B), indicating impaired function in Sertoli cells in these fish. The divergent expression patterns of the germ cell markers, *dazl*, *vasa*, *dnd*, *nanos1*, *nanos2*, *piwil1*, and *piwil2*, were observed in the fish of different genotypes, except *cyp17a1*-/-;*ar*-/-;*npgr*-/- fish (Figure 6C), which might indicate that the upregulated expression of germ cell markers was not critical for the restoration of the defective phenotypes of the testis organization and spermatogenesis resulting from the efficiency of androgen signaling in these various mutant lines (Tang et al., 2018). The Insl3 receptors, *rxfp2a* and *rxfp2b*, expressed in type A spermatogonia and Sertoli cells/myoid cells, respectively, were also decreased in the *cyp17a1*-/-;*ar*-/-;*npgr*-/- fish compared to control males (Figure 6B and C), suggesting compromised Insl3 signaling in *cyp17a1*-/-;*ar*-/-;*npgr*-/- fish. The upregulated expression of the retinoic acid-degrading enzyme, *cyp26a1*, decreased GSI, disrupted testis morphology in the *cyp17a1*-/-;*ar*-/-;*npgr*-/- fish (Figure 6D, 2H, and I) correlated with the observations in the *insl3*-/- males (Crespo et al., 2021).

2) In zebrafish, both Fsh and Lh can signal through the Fsh receptor. Both Fsh and Lh levels are elevated in the crp17a1 mutant fish. Therefore, the crp17a1/ar/fsh triple mutant data is insufficient. The Fsh receptor instead of Fsh β subunit should have been removed in the triple mutant analysis.

It has been found that the hypertrophic testicular development and over-activated spermatogenesis in our *cyp17a1*-/- and *cyp17a1*-/-;*ar*-/- zebrafish, and both Fsh and Lh levels are elevated in these mutant fish (Zhai et al., 2018, PMID: 30202919 and the present study). So, it is needed to figure out the exact course of this phenotype whether is due to elevated Fsh or Lh. We also agree with the fact reported previously that both Fsh and Lh can signal through the Fsh receptor (Zhang et al., 2015, PMID: 25396299 and Xie et al., 2017, PMID: 28611209). That is why the *cyp17a1*-/-;*fsh*-/- double KO fish (Zhai et al., 2018), and *cyp17a1*-/-;*ar*-/-;*fsh*-/- triple KO mutant fish (Figure 6—figure supplement 1) have been generated. In both our double KO and triple KO mutants, the Fsh receptor are still presented, which indicated that Fsh-Fsh Receptor signaling and probably Fsh-Lh receptor signaling are missing in these mutants, while Lh-Fsh Receptor signaling or/and Lh-Lh receptor signaling are still present in these double or triple KO mutants. However, the phenotype of the hypertrophic testicular development and over-activated spermatogenesis seen in the *cyp17a1*-/- or *cyp17a1*-/-;*ar*-/- mutant has been successfully rescued in the *cyp17a1*-/-;*fsh*-/- double KO fish (Zhai et al., 2018), and *cyp17a1*-/-;*ar*-/-;*fsh*-/- triple KO mutant fish (Figure 6—figure supplement 1). That is sufficient to demonstrated that it is the course of the elevated Fsh levels responsible for the phenotype of the hypertrophic testicular development and over-activated spermatogenesis seen in the *cyp17a1*-/- or *cyp17a1*-/-;*ar*-/- mutant, but not because of the elevated levels of Lh signaling, no matter through Lh receptor or Fsh Receptor, which are still presented in our mutant fish. Therefore, to clarify the role of the elevated levels of Fsh or the elevated levels of Lh on the phenotype, we think that the removal of Fsh from the *cyp17a1*-/-;*ar*-/- fish is sufficient and necessary. On the contrary, the depletion of Fshr would not dissect the role of the elevated levels of Fsh through Fsh-Lh receptor signaling or elevated Lh though Lh-Fsh receptor signaling, as they remain.

3) A key issues raised is the reported P and DHP levels and the steroid assays used. One reviewer felt that the reported progestin concentration may be too low to activate the Ar.

For the concentration of the DHP administration, we apologize for the typo error of the concentration description (0.1 µg/L), which should be 0.067 µg/mL. We have corrected this in the revised manuscript (Line 441-446).

Another suggested to measure P and DHP levels using LC/MSMS, which is preferable when measuring P and DHP in this low concentration range as in zebrafish. Whole body extracts contain several compounds co-extracted with steroids that can disturb the assays. The extraction process will lose some P and DHP. If the authors choose to use the same kit, the approach used to extract steroids from tissue samples needs to be specified and quality control experiments/data should be included.

Thanks for the suggestion! In the revised manuscript, the whole-body 11-KT, DHP, and P4 of the control males and *cyp17a1*-/- fish at 3, 3.5, and 4 mpf were measured using Ultra Performance Liquid Chromatography Tandem Mass Spectrometry (UPLC-MS/MS) (Line 133-141). Compared with their control male siblings at their corresponding stages, a significant decrease in 11-KT but an increase in DHP and P4 were observed in the *cyp17a1*-/- fish (Figure 1A-C). The whole-body 11-KT, DHP, and P4 of the control males and *ar*-/- males at 3 mpf were also evaluated, but no significant difference was observed (Figure 1—figure supplement 2). Using the UPLC-MS/MS, we observed that the *ar*-/- males after DHP administration showed a more than threefold concentration of DHP than those reared in the system water (*ar*-/- males reared in system water: 166.1 ± 70.46, *n* = 5; *ar*-/- males reared in DHP: 578.8 ± 379.6, *n* = 5) (Figure 2K).

4) The authors showed that P4 and DHP treatment increased GSI and spermatozoa number. It is pointed out that the doses used in P and DHP treatments may be too low to be effective or the P and DHP quantification may be way off. It would also be important to assess the P4 and DHP concentrations in pharmacologically treated fish. Did they look at the mature sperm number, motility and fertility?

We appreciate the reviewer for pointing out this. For the concentration of the DHP administration, we apologize for the typo error of the concentration description (0.1 µg/L), which should be 0.067 µg/mL. We have corrected this in the revised manuscript (Line 441-446).

As is shown in the UPLC-MS/MS results, *ar*-/- males after DHP administration showed a more than threefold concentration of DHP than those reared in the system water (*ar*-/- males reared in system water: 166.1 ± 70.46, *n* = 5; *ar*-/- males reared in DHP: 578.8 ± 379.6, *n* = 5) (Figure 2K).

We feel sorry for not having the sperm analysis system; instead, we performed fertility (sperm capacity) analysis to assess the sperm fertility of the *ar*+/+ males and *ar*-/- males after treatment with DHP (Line 363-378). Unfortunately, both experiments showed impaired fertility after DHP treatment via artificial fecundation assays with WT females. Based on our observations and several previous publications, the chemical reagents of sex steroids used in the animal administration would result in developmental abnormalities, such as the E2 usually leads to abnormal body growth, enlarged abdomen and liver, and a large amount of liquid in the peritoneal cavity induced by 10 nM E2 treatment from 20 dpf to 40 dpf (Chen et al., 2017), and 10 μg/L (36.71 nM) E2 treatment from 18 dpf to 90 dpf, as well as DHP treatment in the present study (data not shown). On the other hand, based on the results in Figure 1B, the dynamic levels of the elevated DHP seen in the *cyp17a1*-/- fish might suggest that the failure of rescued fertility due to the mismatch with the precise in vivo DHP dynamic levels, or potential side effects of the constant levels of exogenous DHP exposure.

5) Relevant in normal physiology. It would have been useful to compare the presented directly with npgr-/- mutants throughout and compare the cyp17a1-/-; npgr-/- to the ar-/-; npgr-/- and the single mutants. This plus treatment would have clearly answered if progestins play a role in normal physiology. Furthermore, is it possible to inhibit progestin synthesis/secretion in cyp17a1-/- fish or use progestin antagonists (e.g., mifepristone, ORG3170 etc.?) If possible, it will provide further evidence that the elevated levels of progestins is indeed critical.

According to this suggestion, we have analyzed the fish of the eight genotypes from the offspring of the crossed *cyp17a1*+/-;*ar*+/-;*npgr*+/- fish, including *cyp17a1*+/+;*ar*+/+;*npgr*+/+ males (control males), *ar*-/- males, *cyp17a1*-/- fish, *npgr*-/- males, *cyp17a1*-/-;*ar*-/- fish, *cyp17a1*-/-;*npgr*-/- fish, *ar*-/-;*npgr*-/- males, and *cyp17a1*-/-;*ar*-/-;*npgr*-/- fish (Line 183-186).

According to this suggestion, we have performed the anatomical examination and histological analyses (Line 183-198), and gene expression analyses (Line 215-232) in the fish of the eight genotypes from the offspring of the crossed *cyp17a1*+/-;*ar*+/-;*npgr*+/- fish. Then, the testis gene expression levels in different genotypes were also discussed (Line 333-362) (Please see the response to essential revision 1).

Actually, before the submission of the manuscript, we have administrated the *cyp17a1*-/-;*ar*-/- fish with 1 µM mifepristone (RU486) from 60 dpf to 90 dpf. However, no obvious effect on testis organization and spermatogenesis impairment was observed. We think it is possible that the mifepristone treatment for the inhibitory effect of progestin signaling may be concentration-, stage-dependent or the combination of both. On the other hand, though the progestins concentration in the *cyp17a1*-/- fish is higher than that of its control siblings, it varies at different stages (Figure 1B and C). Therefore, a series administrations of mifepristone on *cyp17a1*-/-;*ar*-/- fish at different concentrations or different stages still need further investigation. It also could not be excluded that though mifepristone has been reported as an antagonist for progestin receptor, the inhibitory effect on DHP may not be observed, as it has been reported that RU486 did not block DHP-induced oocyte maturation and *adamts9* expression in zebrafish (Liu et al., 2018, PMID: 30279677).

6) Histological analysis: Figures 2, 3 and 4 show quantitative data (the ordinates are labelled with, "number of sperm per cyst"). However, it is not clear how the quantitative histological data were obtained and what exactly they show.

The quantitation of the data was performed with Image J software, and the details were provided in the Materials and methods section of the revised manuscript (Line 436-440). Briefly, the area in each integrative lumen of seminiferous tubules containing spermatozoa was cut and saved as a new image file for further analysis. After the image was re-loaded, the image was transferred to 8-bit gray, and the threshold was selected for sperm selection. The particle number was analyzed after the watershed under the binary submenu was used to dissect the stacked sperm.

7) Immunofluorescence staining: The authors used an antibody against mouse vasa protein on zebrafish testis sections. The antibody used has not been characterized. The ICC procedure has not been described. No proof is provided that the antibody against a mammalian protein reliably detects zebrafish vasa protein.

As the reviewer doubt whether the antibody we previously used (3008, DIA-AN, Wuhan, Hubei, China) is specific in zebrafish, the immunofluorescence staining with the Vasa antibody purchased from Genetex company (GTX128306-S, United States of America) were performed (which has been identified workability for the immunofluorescence staining in zebrafish) (Zhu et al., 2019, PMID: 31533925).

The ICC procedure has been provided in the Materials and methods (Line 447-454). Immunofluorescence staining was performed using an anti-Vasa rabbit polyclonal antibody (GTX128306-S, GeneTex, Irvine, CA, USA) as the primary antibody (Zhu et al., 2019). Fluorescein (FITC)-conjugated goat anti-rabbit IgG (H^+^L) was used as the secondary antibody (SA00003, Proteintech, Rosemont, IL). As previously described, zebrafish testes were fixed, embedded, sectioned, and stained using standard protocols (Zhu et al., 2019). Nuclear DNA was stained with 4',6-diamidino-2-phenylindole (DAPI). Sections were visualized using 40× objective lenses of an NOL-LSM 710 microscope (Carl Zeiss, Germany). Scale bars are provided for each image.

From the results of immunofluorescence staining, the similar pattern of the Vasa immunofluorescence staining was observed (Line 204-213): the Vasa immunofluorescence localization showed a reduced number of spermatozoa in the lumen of seminiferous tubules of *ar*-/- males (Figure 5D-F), but an increased number of spermatozoa could be observed in the lumen of seminiferous tubules of *cyp17a1*-/- fish (Figure 5G-I) and *cyp17a1*-/-;*ar*-/- fish (Figure 5J-L), again supporting that the reduced number of spermatozoa in the lumen of seminiferous tubules of *ar*-/- males was rescued by the further *cyp17a1* knockout (in the *cyp17a1*-/-;*ar*-/- fish). However, the restoration failed in the *cyp17a1*-/-;*ar*-/- fish with further depletion of *npgr* (in the *cyp17a1*-/-;*ar*-/-;*npgr*-/- fish) (Figure 5M-O), as evidenced by the greater spermatogonia, primary spermatocytes and secondary spermatocytes, but fewer spermatozoa filling in the lumen of seminiferous tubules of *cyp17a1*-/-;*ar*-/-;*npgr*-/- fish (Figure 5O) and the *ar*-/- males (Figure 5F).

In addition, compared with the *cyp17a1*-/- fish and *cyp17a1*-/-;*ar*-/- fish, the observations of greater spermatogonia and spermatocytes, but fewer spermatozoa in the *ar*-/- males, *ar*-/-;*npgr*-/- males, and *cyp17a1*-/-;*ar*-/-;*npgr*-/- fish were observed in the histological analyses of testes (Figure 4J-Q) (Line 194-197).

8) RNA extraction and qPCR: The authors report to have used a single 'housekeeping gene' (ef1a). However, the cellular composition of testis tissue varies in the different mutants, so that it cannot be excluded that also the readings for ef1a changed, depending on the genotype.

According to the suggestion, transcriptome analyses and qPCR verification of the testes of the control males, *ar*-/- males, *cyp17a1*-/- fish, *npgr*-/- males, *cyp17a1*-/-;*ar*-/- fish, *cyp17a1*-/-;*npgr*-/- fish, *ar*-/-;*npgr*-/- males, and *cyp17a1*-/-;*ar*-/-;*npgr*-/- fish, were performed (Line 215-232).

The primers for *ef1a* were referenced from the previous publication (Crespo et al., 2021, PMID: 33589679) (Table 1). We kept using the *ef1a* as the internal reference, as the expressions of *hsd3β1*, *insl3*, *igf3*, *dnd* and *cyp26a1* of qPCR with *ef1a* as the internal reference perfectly matched the expression pattern of the transcriptome analysis (Figure 6E-I). In fact, the qPCR of gonads with *ef1a* as the internal reference is common in the field (Tang et al., 2018, PMID: 29228103. Crespo et al., 2021, PMID: 33589679. Wu et al., 2020, PMID: 32001440. Lu et al., 2017, PMID: 28398516). On the contrary, the expression of *β-actin* in the testes of the fish with the eight genotypes showed more differentiated expression patterns than *ef1a* (Please see Author response image 1).

**Author response image 1. sa2fig1:** 

9) Statistical analysis: The statistical analyses appear incomplete and may even be flawed. This makes it difficult to assess reproducibility of the quantitative measurements, the claims and conclusions made in this study. The authors should consult a statistician to resolve this issue.

Thanks for the pointing out this. In the revised manuscript, the statistical analysis was clarified and performed (modified) as follows: the statistical significance of differences was determined using Student’s *t*-test for paired comparisons and one-way ANOVA, followed by Fisher’s LSD test for multiple comparisons. For all statistical analyses, *P* < 0.05 indicated a significant difference. Significant differences marked with asterisks were analyzed using Student’s *t*-test for paired comparisons, and letters were analyzed using one-way ANOVA, followed by Fisher’s LSD test for multiple comparisons (Line 475-480) (Figures 1, 2, 3, 4 and 6).

10) Titles: It is surprising that the authors repeatedly mentioned "testicular development", including in the title, since the data presented covers adult fish only.

We found that testes of *cyp17a1*-/-;*ar*-/-;*npgr*-/- fish were structurally disorganized with impaired seminiferous tubules and significantly decreased numbers of mature spermatozoa, similar with the previous observations in *ar*-/- males (Crowder et al., 2017. PMID: 29272351) and *cyp11a2*-/- males (Li et al., 2020. PMID: 31693487). According to this suggestion, we have changed "testicular development" to "testis organization" in the revised manuscript, as referred to the previous study (Crowder et al., 2017. PMID: 29272351. Li et al., 2020. PMID: 31693487).

11) A number of substantive concerns were raised with the statements/overstatements in the text. These should be addressed.

We are sorry for the minor concerns. The revised manuscript has been edited with Elsevier Webshop and double checked by us

Reviewer #1:The authors were intrigued by the observation that loss of the androgen receptor (ar) in zebrafish has a clear impact on spermatogenesis while spermatogenesis remained intact after blocking androgen production through loss of the enzyme cyp17a1. This supports an earlier conclusion also proposed by others: in fish, spermatogenesis can be supported by other than androgen-driven signaling pathways. Since the authors found progestin levels to be higher in cyp17a1 mutants, and since results published previously by others showed that progestins stimulated different aspects of spermatogenesis (spermatogonial development, meiosis, sperm capacitation) in different fish species (eel, tilapia, trout, zebrafish), the authors asked if removing also the nuclear progesterone receptor (npr) from cyp17a1 mutants would result in a spermatogenesis phenotype in zebrafish after all. To answer this valid question, the authors generated mutant lines to seek support for the hypothesis that progestin signaling via its nuclear receptor can maintain spermatogenesis in androgen-depleted cyp17a1 mutants. Also ar-KO zebrafish were subjected to progesterone treatment, although the reasons for that were less clear.

Based on our previous study (Zhai et al., 2018, PMID: 30202919), we hypothesize that the accumulated progestin, DHP, which is upstream product of the androgen, due to *cyp17a1* knockout, may contributed to the normally developed testis, as DHP has been reported as the major, potent, and biologically relevant progestin in teleosts (Wang et al., 2016, PMID: 27113852. Scott, 2010, PMID: 20738705). Therefore, the 11-KT, DHP, and P4 measurements in *cyp17a1*-/- fish and *ar*-/- fish, as well as their control siblings were performed (Figure 1 and Figure 1—figure supplement 2). The assay of the DHP administration of *ar*-/- males (Figure 2) is based on Figure 1 and Figure 1—figure supplement 2. This has been clarified in Line 147-149.

Surprisingly, the mutant most pertinent to the main question, the cyp17a1/npr double mutant, was only used to analyze the expression of selected testicular genes. No anatomical or histological data were included, in contrast to all other lines, thus leaving the main question unanswered regarding effects on spermatogenesis.

Thanks for the suggestion. According to this suggestion, we have analyzed the fish of the eight genotypes from the offspring of the crossed *cyp17a1*+/-;*ar*+/-;*npgr*+/- fish, including *cyp17a1*+/+;*ar*+/+;*npgr*+/+ males (control males), *ar*-/- males, *cyp17a1*-/- fish, *npgr*-/- males, *cyp17a1*-/-;*ar*-/- fish, *cyp17a1*-/-;*npgr*-/- fish, *ar*-/-;*npgr*-/- males, and *cyp17a1*-/-;*ar*-/-;*npgr*-/- fish. Please see the response to essential revision 1.

Instead, triple mutants were generated by removing also the androgen receptor (cyp17a1/npr/ar) or the Fsh β subunit (cyp17a1/npr/fshb). Unfortunately, the loss of a third gene does not allow to draw "clean" conclusions regarding the consequences of the loss of npr in males unable to produce androgens; also, it is not clear what led to the choice of ar or fshb as third genes to be removed. These points are further discussed in the following.

To our knowledge, compared with the knockdown or chemical reagents administration, the knockout strategy is the most potent method to draw the conclusions. And we would point out that the third gene we knockout from *cyp17a1*-/-;*ar*-/- fish is *npgr* and *fshb* respectively, not *ar* or *fshβ* from *cyp17a1*-/-;*npgr*-/- fish as reviewer mentioned. And the reason we choose to knockout *npgr* or *fshb* on *cyp17a1*-/-;*ar*-/- fish is based on the accumulated DHP and upregulated expression of the pituitary gonadotropin *fshb* in the *cyp17a1*-/- fish, these have been given clearly (Figure 1 and Line 179-180 and 313-324).

Follicle-stimulating hormone β subunit (fshb)Referring to works published by others, the authors state correctly that Fsh can stimulate spermatogenesis independent of sex steroid signaling. Since elevated levels of both Lh and Fsh have been reported in cyp17a1 KO zebrafish, the authors decided to remove the Fsh β subunit gene (reminiscent of work in their 2018 paper), attempting to study the contribution, if any, of Fsh signaling to maintaining spermatogenesis. However, in the cyp17a1/ar/fshb triple mutant, the Fsh receptor is still present. Since the also elevated Lh levels can cross-activate the Fsh receptor (Xie et al., 2017, DOI: 10.1530/JOE-17-0079), it is conceivable that Fsh receptor-mediated signaling remained relevant. Therefore, it seems that removal of the receptor, instead of the ligand, would have been the option to choose for investigating the role of Fsh signaling.

Please see response to essential revision 2.

The authors concluded that removing fshb had no effect on spermatogenesis, referring to GSI levels being similar to controls (L290). However, the authors did not point out that in the cyp17a1/ar/fshb triple mutant, GSI values were halved compared to the cyp17a1/ar double mutant, i.e. removal of fshb apparently did have an effect.

The statement that upregulated Fshβ contributed to the hypertrophic testicular development and over-activated spermatogenesis in the *cyp17a1*-/-;*ar*-/- zebrafish given by us it is certainly based on the facts that *cyp17a1*/*ar*/*fshb* triple mutant showed decreased GSI compared to the *cyp17a1*/*ar* double mutant (Line 318-324).

Before the generation of the *cyp17a1*/*ar*/*fshb* triple mutant and the analysis, the Fshb contribution in testis organization or spermatogenesis of *cyp17a1*-/-;*ar*-/- could not be concluded. For example, unlike *fshβ*, the knockout of *lhβ* has not any obvious restoration on hypertrophic testis or enhanced spermatogenesis in the *cyp17a1*-/- fish (Data not shown). Therefore, the dissection of the upregulated *lhβ* and *fshβ* in our generated mutants based on these genetic data is very important and could not be assumed, as we could not make the conclusion (the upregulated Fshβ contributed to the hypertrophic testicular development and over-activated spermatogenesis in the *cyp17a1*-/-;*ar*-/- zebrafish) until we actually further depleted *fshβ* from the *cyp17a1*/*ar* double mutant and analyzed.

Androgen receptorThe authors mention the possibility that elevated levels of low affinity ligands might activate the Ar in the absence of androgens, such as the elevated progestin levels of found in cyp17a1 mutants. Based on the data presented in Figure 1, the reviewer calculated the combined concentrations of P and DHP to be ~60 nM (data from Figure 1). This is 10- to 20-fold lower than the progestin concentration required to induce some activity of the zebrafish Ar; moreover, the progestin-induced Ar activity is only ~1/10th of the activity induced by androgens (based on the data cited by the authors as de Waal et al., 2008). Hence, available information suggested that progestin concentrations were too low and could only marginally activate the Ar, and therefore seem irrelevant for cross-activating the zebrafish Ar. The reviewer wonders about potential other reasons for carrying out the experiments shown in Figure 2.

We would remind that the affinity between progestin and Ar was not relevant in the present study, as we think that the accumulated progestin in the *cyp17a1*-/- fish and *cyp17a1*-/-;*ar*-/- fish execute its function with nPgr. Thus, the concentration of the progestins, the reviewer think that is too low (~60 nM) to activate Ar is not related, as we are trying to say throughout the manuscript is that nPgr is the receptor of accumulated progestin in facilitating testis organization and spermatogenesis, which is independent from androgen signaling. Moreover, though the reviewer think that the whole-body progestin concentration is too low, we would remind that it could not reflect the concentration status of progestins in the gonads as well.

A question requiring attention is the treatment with progestins (Figure 2) also relating to Figure 1. It shows the endogenous progestin levels in ar-KO mutants being ~5-10ng/g = 5-10µg/kg, equivalent to 5-10µg/L. In Figure 2 and the associated text, the authors report clear effects following exposure of ar-KO fish to P or DHP at a concentration of 0.1µg/L. Reporting clear treatment effects using progestin concentrations 50-100 times lower than the endogenous concentrations would require a careful discussion.

We appreciate the reviewer for pointing out this. We apologize for the typo error of the progestin concentration description (0.1 µg/L), which should be 0.067 µg/mL for the administration of fish from 50 to 90 dpf. We have corrected the mistake in the revised version of the manuscript (Line 444).

Other results obtained with fish carrying a mutated ar await being discussed. For example, the authors find, similar to others, that spermatogenesis is compromised in ar-KO zebrafish and claim (L193 and following) that testicular gene expression analysis (Figure 6) supports this finding. However, the five 5 germ cell genes quantified seem clearly elevated in ar-KO fish showing compromised spermatogenesis after loss of the ar. How to reconcile increased germ cell gene expression with compromised spermatogenesis remains unclear. Similarly, germ cell gene expression is reported to be upregulated in the cyp17a1/npr/ar triple mutants compared to cyp17a1/npr double mutant, i.e. the combination of impaired spermatogenesis and increased germ cell gene expression following removal of the ar gene comes up again. This time, however, not all five germ cell genes but only four were responding, and it is unclear why is there an "odd one out" in this specific genetic model.

Please see the response to essential revision 1.

More comprehensive discussion is perhaps also required regarding the Tgf β family member Amh. The authors did discuss the role of Amh in spermatogonial differentiation, but may have overlooked Amh-promoted transition into meiosis (https://doi.org/10.1016/j.mce.2020.110963).

Based on the transcriptome analyses and qPCR verification, the gene expression analyses have been re-written in the revised manuscript (Line 215-232, 333-362). Please see the response to essential revision 1.

The authors repeatedly make the point, including in the title, that progestin signaling stimulates spermatogenesis independently from androgen signaling. This wording is not sufficiently precise. The genetic evidence presented is based on fish that are unable to produce androgens. This does not exclude the possibility that the Ar protein can exert biological activity in the absence of androgens (or any other type of ligand). Ligand-independent Ar action may indeed be related to changes in gene expression induced by removing the ar from the cyp17a1/npr double mutants (see above). Therefore, the conclusion/point (progestin stimulates spermatogenesis independently from androgen signaling) should be restricted to fish unable to produce androgens. Another reason to do so is that in the presence of androgens (the physiological situation), androgen/Ar-mediated effects may very well interact with progestin/Npr effects on spermatogenesis.

Though Ar has ligand-independent action, *i.e.*, Ar protein can exert biological activity in the absence of androgens (or any other type of ligand), the following evidence support the hypothesis that the *cyp17a1* knockout activated the compensatory pathway in promoting testis organization and spermatogenesis, which is *ar*-independent, but *npgr*-dependent: first, the testis organization and spermatogenesis of *ar*-/- males were evidently rescued by DHP administration (Figure 2). Second, the enhanced spermatogenesis could be achieved in *cyp17a1*-/-;*ar*-/- zebrafish, which clearly indicates that this phenotype of the enhanced spermatogenesis can be independent from the presence of the androgen/ar signaling in zebrafish (Figure 3). Third, compared with the *cyp17a1*-/- fish and *cyp17a1*-/-;*ar*-/- fish, the GSI and spermatozoa number were both significantly declined in *cyp17a1*-/-;*npgr*-/- fish (Figure 4). Four, compared with the *cyp17a1*-/-;*ar*-/- fish, the testis organization and spermatogenesis failed in the *cyp17a1*-/-;*ar*-/-;*npgr*-/- fish (Figure 4). Besides, no significantly differences between the levels of the specific gene markers of the testis samples from the *cyp17a1*-/- fish and *cyp17a1*-/-;*ar*-/- fish (Figure 6), which suggesting no evident effects of ligand-independent Ar action has been seen in the course of the spermatogenesis.

Considering the comments raised by the reviewer, to emphasize that the statement is based on the premise of accumulated progestin in the *cyp17a1*-/- fish, we have changed the title to "High level of progestin signaling facilitates testis organization and spermatogenesis independently from androgen signaling in fish".

A repeatedly occurring statement requiring attention is the mentioning of testis development or testicular development in the manuscript. It would be appropriate to remove all of this from the manuscript, including the title, because respective data is not presented. The datasets in the manuscript refer exclusively to adult males.

Please see the response to essential revision 10.

Another point refers to an alternative understanding of the phenotype of the cyp17a1 mutant. First mentioned in L96-97, this mutant is described as presenting stimulated or enhanced spermatogenesis. The reviewer assumes that the authors refer to the GSI levels and spermatozoa number being higher than in WT adults older than 3 months. The age is relevant since in young adults up to 3 months of age, the GSI is still similar to WT controls. However, neither in the previous study (Zhai et al., 2018) nor in the present manuscript, spermatogenic activity has been studied in detail. The fact that the phenotype of increased GSI/sperm number developed with time after 3 months of age (as described in Zhai et al., 2018) and the histological pictures shown in Zhai et al., 2018 and in the present manuscript, suggests that the weight gain is based on progressively accumulating spermatozoa. Since cyp17a1 mutants don't show reproductive behavior/spawning, the reviewer understands the slowly increasing GSI by progressively accumulating sperm formed at a normal rate. Claiming enhanced spermatogenesis requires research into the dynamics of this cellular development that is missing as yet.The accumulation of sperm may also explain increased DHP levels. The enzyme converting 17aP to DHP, 20bHSD, is highly active in spermatozoa of salmonid and cyprinid species, including the zebrafish relatives carp and goldfish. In this regard, elevated DHP levels may reflect an "accidental side-effect" of sperm with their associated 20bHSD activity, accumulating in the testes of cyp17a1 mutants, and then metabolizing the substrate 17aP that is available in high levels due to the loss of cyp17a1.

We would remind that the increased GSI and stimulated spermatogenesis, as well as normal fertility (sperm capacity) as evidenced with artificial fecundation, is common in the field. As histological analysis, anatomical observation, germ cell distribution analysis, etc. are common in the field (Crowder et al., 2018; Lau et al., 2016; Li et al., 2020; Lu et al., 2017; Tang et al., 2018; Tang et al., 2016; Yin et al., 2017; Yu et al., 2018; Zhai et al., 2018; Zhu et al., 2015). On the contrary, to our knowledge, the spermatogenic activity mentioned by the reviewer is rarely used in zebrafish.

In fact, only the males of *cyp17a1*+/+, *cyp17a1*+/- and *cyp17a1*-/- zebrafish were kept in the fish systems for the further analyses at 3 mpf. Thus, there is no difference of the reproductive behavior/spawning of the test male fish between different genotype fish from 3 mpf. It should not be related with the accumulating sperms during the 3 mpf to 6 mpf.

In addition, it has been reported that the elevated levels of *fshβ* transcriptional expression if responsible for the increased GSI and spermatogenesis in our early report (Zhai et al., 2018). We would remind that at 3 mpf, the *cyp17a1*-/- fish has increased GSI, though the statistical difference is not significant (Zhai et al., 2018, Figure 1K). On the other hand, the levels of *fshβ* transcriptional expression kept increasing during the 3 to 6 mpf (Zhai et al., 2018, Figure 6F).

Finally, some of our unpublished data also provide evidence indicating the promotion function of Fshβ on differentiation of the spermatozoa at early stage (at 45 dpf, before the onset of the puberty in WT male zebrafish), the spermatozoa could be observed in the testis of *cyp17a1*-/- fish, which could not be observed in the control males until 50-55 dpf. More specifically, the accelerated spermatogenesis onset of *cyp17a1*-/- fish was rescued when *fshb* is removed (in the *cyp17a1*-/-;*fshb*-/- fish) (Please see Author response image 2).

Overall, the reviewer considers it as a pity that the manuscript does not report also first results regarding mechanisms underlying the progestin-stimulated maintenance of spermatogenesis in cyp17a1 KO zebrafish. Such data would elevate the manuscript from its descriptive nature. Nevertheless, the authors have generated valuable genetic models that may allow, in the future, not only to provide the previously reported, progestin-mediated stimulation of spermatogenesis with a firm genetic basis in zebrafish, but also to broaden our knowledge on the mechanisms underlying progestin effects on spermatogenesis.

We agree that we did not reveal the mechanisms underlying the progestin-mediated promotion of spermatogenesis in *cyp17a1* KO zebrafish in the present study. However, we provide the evidence for the capacity of the progestin-mediated promotion of spermatogenesis, and which can be effective independent from the androgen/androgen receptor signaling. The mechanism would be elucidated in further studies.

Materials and methods sectionSteroid assays: The approach used to extract steroids from tissue samples needs to be specified and quality control experiments/data are missing. Contrary to the authors' statement, the procedure is not specified by the manufacturer. Looking up the manufacturer's specifications showed that these assays are meant for aqueous samples (plasma, serum, whole blood, urine samples, in vitro medium samples are mentioned, but not tissue or whole body extracts). Therefore, validation of the techniques for a new type of samples is required. This also includes the point that the authors apparently did not control for procedural losses. In addition, whole body extracts contain several compounds co-extracted with steroids that can disturb the assays, as indicated by the manufacturer, who suggested to remove the impurities even from much less complex samples such as blood plasma extracts, before introducing the samples into the assay. Has this been done? Finally, the authors do not report validation and standardization of the assays (e.g. extraction efficiency and potential correction for procedural losses, or reliability [i.e. is a spiked amount of steroid found back at the expected concentration?]). As reported now, results on steroid quantifications cannot be accepted as sufficiently reliable or accurate.

Thanks for the suggestion. In the revised manuscript, the whole-body 11-KT, DHP, and P4 of the control males and *cyp17a1*-/- fish at 3, 3.5, and 4 mpf were measured using Ultra Performance Liquid Chromatography Tandem Mass Spectrometry (UPLC-MS/MS) (Line 133-141). Compared with their control male siblings at their corresponding stages, a significant decrease in 11-KT but an increase in DHP and P4 were observed in the *cyp17a1*-/- fish (Figure 1A-C). The whole-body 11-KT, DHP, and P4 of the control males and *ar*-/- males at 3 mpf were also evaluated, but no significant difference was observed (Figure 1—figure supplement 2). These results not only confirmed that androgen biosynthesis is impaired in the *cyp17a1*-/- fish, but also reminded us that compared with the *cyp17a1*-/- males, which demonstrated higher concentrations of DHP and P4 than its control male siblings, the *ar*-/- males exhibited comparable concentrations of DHP and P4 with its control male siblings.

Histological analysis: Figures 2, 3 and 4 show quantitative data (the ordinates are labelled with, "number of sperm per cyst"). However, the respective M and M section does not detail how the data was obtained and processed. Regarding the label on the ordinate, please refer to literature on spermatogenesis in fish, the point being that after completion of spermiogenesis, the spermatogenic cysts open and germ cells are released by the Sertoli cells into the lumen of the spermatogenic tubules. From then on, the germ cells are called spermatozoa, which are no longer in cysts. Hence, "sperm per cyst" is a contradiction in terms, leading to the question if "sperm" or "per cyst" is not correct? These points require clarification for the data to be kept in the manuscript. Also the legend to Figure 5 requires attention in this regard. The authors state that scale bars are provided in each image – this is not correct, since for example on Figure 2, there are no scales; the scales on Figure 4, on the other hand, are probably not correct. In L342, please replace size by thickness.

Please see response to essential revision 4.

Thanks for pointing out this. We have changed "spermatogenic cyst" to "lumen of seminiferous tubules", which has been reported contains developing sperm surrounding mature sperm (Crowder et al., 2017).

Following the suggestions, the scale bars in each image has been added in the revised version of the manuscript (Figure 2-5). Besides, the "size" has been replaced with "thickness" (Line 433).

P4 and DHP treatment: Please refer to the point made above in this context in Public Review (100-fold lower concentration used for treatments than found in the fish).

For the concentration of the DHP administration, we apologize for the typo error of the concentration description (0.1 µg/L), which should be 0.067 µg/mL. We have corrected this in the revised manuscript (Line 441-446).

Immunoflourescence staining: The authors report to have used an antibody against mouse vasa protein on zebrafish testis sections. The antibody used has not been characterized. The ICC procedure has not been described. No proof is provided that the antibody against a mammalian protein reliably detects zebrafish vasa protein. In the Results section, the authors state in L182 that the ICC data demonstrate "…a reduced number of spermatozoa…in ar-/- males…but increased number of spermatozoa … in … cyp17a1 fish". However, spermatozoa do not express vasa protein. Moreover, since the ICC results in Figure 5 have neither been quantified nor normalized, statements referring to differences in cell numbers should be avoided.

Please see response to essential revision 7.

We agree that spermatozoa do not express vasa protein; therefore, this was utilized for distinguishing the SZ from SG and SC cells (Line 201-203).

We described the cell with "greater" in the manuscript, not mentioned with "statistically difference". Please note that this is common in the field, especially in some special situations that it is hard to perform the statistical analysis (Crowder et al., 2018; Tang et al., 2018; Yu et al., 2018; Zhang et al., 2020).

RNA extraction and qPCR: The authors report to have used a single 'housekeeping gene' (ef1a). However, the cellular composition of testis tissue varies in the different mutants, so that it cannot be excluded that also the readings for ef1a changed, depending on the genotype. The request is to include the ef1a data as a separate graph in Figure 6 (e.g. as Figure 6K) for the different genotypes and subject them to statistical analysis. If differences are detected between genotypes, it is possible to use the approach explained in the following. For the future (or in case ef1a shows changes here), consider using at least three different reference genes, showing a large range of expression levels, and calculate qPCR results using their geometric mean as normalization.

Please see response to essential revision 8.

Statistical analysis: This is one of the more critical points in the msnuscript, since the potential problems here prevent evaluation of all quantitative data presented in the manuscript. The number of replicates per experiment/treatment group has not been provided. It is mentioned that experiments were repeated three times. However, how were the data obtained from these three experiments then used for statistical analysis and plotting of the graphs? In most graphs presenting quantitative data, the authors compare more than 2 groups with each other, usually 4-6 groups. The authors state in L369 that "differences were assessed using Student's t-test." With this test, however, only differences between two groups can be assessed, under the condition that the data is distributed normally. However, neither was data distribution tested, nor were there only two groups. Therefore, all statistical analyses need to be repeated.

Thanks for the pointing out this. In the revised version of the manuscript, the statistical analysis was performed as follows: the statistical significance of differences was determined using Student’s *t*-test for paired comparisons and one-way ANOVA, followed by Fisher’s LSD test for multiple comparisons. For all statistical analyses, *P* < 0.05 indicated a significant difference. Significant differences marked with asterisks were analyzed using Student’s *t*-test for paired comparisons, and letters were analyzed using one-way ANOVA, followed by Fisher’s LSD test for multiple comparisons (Figures 1, 2, 3, 4 and 6, Line 475-480).

IntroductionAs indicated above, this following list mentions selected points independent of the re-analysis of the data. The term "selected" is used also because the number of points requiring comments is high and the list is far from complete. Hence, the authors will have to identify the points not mentioned below during a thorough re-writing process.In the reviewer's opinion, the complete 1st paragraph of the Introduction can be removed.

Following the suggestion, we have re-written the 1st paragraph in our revised manuscript. We think it is necessary to summarize the current information about the knowledge of functions of androgen and progestin in zebrafish revealed with KO models. Especially for the 3rd conclusion, which we think that it is necessary (the reviewer think it should be removed, please see below), as the function of progestin signaling has only be focused on ovulation in female zebrafish previously. We think that the revised 1st paragraph, in which the current information of the androgen and progestin signaling in zebrafish has been introduced, is helpful for understanding the significance of our finding (Line 53-65).

L49-56: This sentence contains several unclear or awkward statements (e.g. "the reproduction research" – grammar; "advantage of gonadal dissection" – ??; "successfully adopted KO strategy" – applies to virtually thousands of species; "pharmacological utilization" – ???).

Following the suggestions, this paragraph has been re-written in our revised manuscript (Line 52-61). As we are not native English speakers, the revised manuscript has been proofread by Elsevier English Language Editing Webshop.

Conclusions in L57-61: the 1st conclusion is irrelevant in the present context; the 2nd conclusion is in part wrong (see following point); the 3rd conclusion is irrelevant in the present context

For the 3rd conclusion, it is necessary as the function of progestin signaling has only be focused on ovulation in female zebrafish previously. Following the suggestions, this paragraph has been re-written in our revised manuscript (Line 61-65).

L58-61: Contrary to the authors' statement androgen signaling is not essential for testicular differentiation and development or for spermatogenesis in fish and the papers cited do not make this claim either. After all, the authors did study testis tissue from ar-KO mutants and this tissue contained all types of germ cells including sperm (e.g. Figure 2K). This wrong statement is repeated in L62 (wrong regarding testis development in fish).

Following the suggestions, Line 58-61 has been re-written in our revised manuscript, and Line 62 has been removed (Line 61-65).

L79-82: A series of statements is made but not supported by citations.

The citations have been provided (Line 88).

L103-104: The authors state that "administration of P4 and DHP effectively restored … GSI and spermatozoa number…". However looking at the graphs in Figure 2 G and N, the formulation is not correct since the controls showed higher levels (pending the re-analysis of the data, significantly higher in the controls). This type of error/overstatement occurs frequently.

We agree that the number of spermatozoa in each lumen of seminiferous tubules of *ar*-/- fish after DHP administration is still significantly lower than those of the control fish (Figure 2J). However, the impaired phenotypes have been significantly improved when compared with the *ar*-/- males reared in system water supplemented with the vehicle. It is hard to rescue the spermatogenesis in zebrafish fully with the in vitro administration of steroid. As seen in the Figure 1B, the in vitro immersion of DHP could be difficult to precisely match the dynamic of progestin production in vivo. On the other hand, based on our observations and several previous publications, the rescue effects of steroids, such as the E2 usually leads to abnormal body growth, enlarged abdomen and liver, and large amount of liquid in the peritoneal cavity induced by 10 nM E2 treatment from 20 dpf to 40 dpf (Chen et al., 2017, PMID: 29055862), and 10 μg/L (36.713 nM) E2 treatment from 18 dpf to 90 dpf (Data not shown in our previous study). E2 have been widely stated for its rescue effects on the ovarian differentiation of the *cyp17a1*-/- fish or *cyp19a1a*-/- fish (Zhai et al., 2018, PMID: 30202919; Lau et al., 2016, PMID: 27876832; Yin et al., 2017, PMID: 28575219), though the fertility could not be successfully restored. Even though, the observations can still support the role of estradiol on ovarian differentiation. Likewisely, we would like to remind that the function of the administration of the DHP in the present study is meaningful. The improvement of the testis organization and spermatogenesis is statistically recovered, which supports our statement, as we have illustrated the statistical significance after statistical analysis in the Figure 2.

L110: Another example for an overstatement is to use the term "arrested" to describe the effect of ar loss on spermatogenesis. While this is correct for mammals, it is not for zebrafish (see the authors' data in Figure 2K, showing an ar-KO testis with all types of germ cells including spermatozoa, i.e. no arrest).

Thanks for the pointing out this. The "arrested" has been replaced with "impaired" throughout the manuscript and highlighted in red.

Overall, the introduction contains passages that rather read like Results or Discussion section. Please amend.

Thanks for the pointing out this. The introduction has been re-organized and re-written (Line 52-119).

Results and DiscussionBoth sections contain statements that would require comments. However, the contents of these statements have been touched upon already, in one way or another, in the points discussed above. Therefore, the reviewer does not make specific comments to these sections, also considering the statistical/data uncertainties and technical questions remaining to be solved, which will likely result in several changes in these two sections.

Thanks for the efforts in reviewing the manuscript and provide the valuable suggestions. We have carefully revised the manuscript and addressed the comments point by point. We hope you will find it ideally suited for the journal.

Reviewer #2:The authors present a paper suggesting that progestin facilitates testicular development in zebrafish. Overall, the paper tries to explain interesting observations made in their cyp17a1-/- lines that are different compared to fish with androgen deficiency and androgen resistance. Interestingly, the specific block in steroidogenesis led to an increase of progestins and the authors work up the hypothesis that progestins represent an androgen-independent regulator of testicular development and spermatogenesis. Is this a compensated "disease" state or a mechanism that is relevant under normal physiological circumstances?The authors present quite some detailed analysis of a structured combination of various mutants/ double/ triple mutants to explore their hypothesis, which appears a major strength of the paper. There some aspects such as the effect pf progestin treatment on wildtypes and the discussion if these are physiological, pathophysiological or pharmacological effects that would need further analysis and clarification. A major weakness of the paper is that no direct comparison with npgr-/- lines have been made to explore the questions not only from the angle of androgen deficiency and resistance with progestin excess.Overall this paper might form the basis some interesting novel research investigating the interplay between androgens and progestins in zebrafish gonadal development and function. The work is convincing, but despite the fact that progestins are known to play an important role in spermatogenesis, it is remains unclear if progestins play the same significant roles in wildtype fish.Despite a number of very interesting experiments that clearly show a role progestins in testicular development in androgen-deficient/ resistant zebrafish, it remains unclear if this is relevant in normal physiology or only to explain overserved phaenomena in mutant lines. It would have been useful to compare the presented directly with npgr-/- mutants throughout and compare the cyp17a1-/-; npgr-/- to the ar-/-; npgr-/- and the single mutants. This plus treatment would have clearly answered if progestins play a role in normal physiology. Importantly, how high are concentrations in the treated zebrafish? Are these physiological or pharmacological effects?

According to this suggestion, we have analyzed the fish of the eight genotypes from the offspring of the crossed *cyp17a1*+/-;*ar*+/-;*npgr*+/- fish, including *cyp17a1*+/+;*ar*+/+;*npgr*+/+ males (control males), *ar*-/- males, *cyp17a1*-/- fish, *npgr*-/- males, *cyp17a1*-/-;*ar*-/- fish, *cyp17a1*-/-;*npgr*-/- fish, *ar*-/-;*npgr*-/- males, and *cyp17a1*-/-;*ar*-/-;*npgr*-/- fish. Please see response to essential revision 1.

As is shown with the UPLC-MS/MS result, the *ar*-/- males after DHP administration showed a more than threefold concentration of DHP than those reared in the system water (*ar*-/- males reared in system water: 166.1 ± 70.46, *n* = 5; *ar*-/- males reared in DHP: 578.8 ± 379.6, *n* = 5) (Figure 2K) (Line 155-157).

The origin thin and transparent testis of *ar*-/- males become white and full-grown after DHP treatment, indicating an improved testis development (Figure 2B, D, E, G, I and J); however, based on our observations, for the *ar*+/+ males with or without DHP treatment, the GSI and sperm number analysis seems largely unaffected (Figure 2A, C, E, F, H and J). This may be the presence of androgen signaling in *ar*+/+ males, which has been reported to be necessary for proper sex differentiation and development in vertebrates.

It would be very interesting to see the effects of synthetic progestins, such as medroxyprogesterone, levonorgestrel, dydrogesterone, etc, on the *ar*-/- males. We are also looking forward to seeing whether these chemicals are capable in rescuing the defective testis organization and spermatogenesis in *ar*-/- males in the further studies. However, as we mentioned in the manuscript (Line 363-378), the in vitro progestins exposure of *ar*-/- males could be difficult to match precisely the in vivo dynamic DHP production of *cyp17a1*-/- fish, and the potential side effects of the constant levels of exogenous progestins exposure might affect body growth, development, or reproduction of *ar*-/- males.

Overall, it appears that the progestin excess does not compensate it rather leads to hyperplastic testes, which in itself appears interesting and could be further addressed. Could this be relevant to answer environmental questions/ exogenous progestin exposure?Lines 253-257: This raises the key question of progestins are really required for testicular developments and spermatogenesis.I disagree with this statement. The authors demonstrate is that progestins can override defective androgen synthesis and the effects on testicular development. It would have been very interesting see if sperm from the various mutant lines can fertilise eggs. In addition, the assessment of mature sperm count and their motility would have been vital to understand the phenotype better.

We have re-written this section (Line 363-378). Please see response to essential revision 4.

If progestins are required in a way that the authors suggest then a direct comparison with the npgr-/- in their representations would be very useful I not essential.

Thanks for the suggestion. We agree that the additional analysis of the other genotypes is helpful; therefore, for the anatomical analysis, histological analysis and gene expression analysis, we have analyzed the fish of the eight genotypes from the offspring of the crossed *cyp17a1*+/-;*ar*+/-;*npgr*+/- fish, *i.e.,* control males, *ar*-/- males, *cyp17a1*-/- fish, *npgr*-/- males, *cyp17a1*-/-;*ar*-/- fish, *cyp17a1*-/-;*npgr*-/-, *ar*-/-;*npgr*-/- males, and *cyp17a1*-/-;*ar*-/-;*npgr*-/- fish (Figures 4 and 6, Line 183-198, 215-232 and 333-362). The comparison results have been provided in our revised manuscript (Figures 4 and 6). Since the progestin level was significant higher in *cyp17a*-/- and *cyp17a*-/-; *ar*-/- males compared with their control males respectively, the facts of the impaired testis organization and spermatogenesis could be caused by further depletion of *npgr* in either *cyp17a1*-/- or *cyp17a*-/-;*ar*-/- background, clearly indicating that the high level of progestins/npgr signaling resulted from *cyp17a1*-deficiency is required for the proper testis organization and spermatogenesis.

It remained unclear why effects on the hypothalamic pituitary gonadal axis have not been assessed? This could provide further evidence about the degree of compensation by progestins. It would have been interesting if there could be synergistic effect of gonadotrophs or if there is a downregulation or even more interestingly if lh and fsh are differentially affected.

The expression analysis of pituitary *fshβ* in fish of the *cyp17a1*+/+;*ar*+/+;*npgr*+/+ males, *ar*-/- males, *cyp17a1*-/- fish, *npgr*-/- males, *cyp17a1*-/-;*ar*-/- fish, *cyp17a1*-/-;*npgr*-/- fish, *ar*-/-;*npgr*-/- males, and *cyp17a1*-/-;*ar*-/-;*npgr*-/- fish were performed. The expression of pituitary *fshβ* in fish of the eight genotypes mentioned above was also examined (Line 324-332). We found that the expression of *fshβ* was upregulated in the *cyp17a1*-/- fish and *cyp17a1*-/-;*ar*-/- fish, as expected based on our previous study (Zhai et al., 2018). The *npgr* depletion did not affect pituitary *fshβ* expression, as *npgr*-/- males and *ar*-/-;*npgr*-/- males exhibited comparable *fshβ* expression to control males; however, the addition of *cyp17a1* depletion significantly upregulated *fshβ* expression in the males of these genotypes (in the *cyp17a1*-/-;*npgr*-/- fish and *cyp17a1*-/-;*ar*-/-;*npgr*-/- fish) (Figure 6—figure supplement 2). From these results, we observed the impaired spermatogenesis and testis organization, and the upregulated expression of pituitary *fshβ* in *cyp17a1*-/-;*ar*-/-;*npgr*-/- fish.

This could be very different to the upregulation of the HPG axis in recently described androgen deficiency fish is secondary. Parts of the discussion are not entirely clear as it cannot be expected to restore testicular development in other species. The message of this part of the discussion appears not entirely clear.

We agree. This paragraph has been re-written. In the revised manuscript, the gonadal-pituitary feedback axis in the *cyp17a1*-/- fish and *cyp19a1a*-/- fish were retained and discussed, to remind the readers that the upregulated Fshβ contributed to the hypertrophic testicular development and over-activated spermatogenesis in the *cyp17a1*-/-;*ar*-/- zebrafish, which were also observed in the *cyp17a1*-/- fish and *cyp19a1a*-/- fish (Figure 6—figure supplement 1 and Line 309-324).

The steroid assays are not well described. How have they been validated? Assays to measure all compound by LC/MSMS are available and preferable when measuring analytes in this low concentration range as in zebrafish. The use of ELISAs for specific hormones appears outdated and less robust than LC/MSMS assay. This appears to be a real weakness of the paper.

Thanks for the suggestion! In the revised manuscript, the whole-body 11-KT, DHP, and P4 of the control males and *cyp17a1*-/- fish at 3, 3.5, and 4 mpf were measured using Ultra Performance Liquid Chromatography Tandem Mass Spectrometry (UPLC-MS/MS) (Line 133-141). Compared with their control male siblings at their corresponding stages, a significant decrease in 11-KT but an increase in DHP and P4 were observed in the *cyp17a1*-/- fish (Figure 1A-C). The whole-body 11-KT, DHP, and P4 of the control males and *ar*-/- males at 3 mpf were also evaluated, but no significant difference was observed (Figure 1—figure supplement 2A-C).

It would be very important to assess the P4 and DHP concentrations in pharmacologically treated fish. The GSI does not seems to change; are the treated animals with larger testes heavier/ longer?

Using the UPLC-MS/MS, we observed that the *ar*-/- males after DHP administration showed a more than threefold concentration of DHP than those reared in the system water (*ar*-/- males reared in system water: 166.1 ± 70.46, *n* = 5; *ar*-/- males reared in DHP: 578.8 ± 379.6, *n* = 5) (Figure 2K) (Line 155-157).

The origin thin and transparent testis of *ar*-/- males become white and full-grown after DHP treatment, indicating an improved testis development (Figure 2B, D, E, G, I and J); however, based on our observations, for the *ar*+/+ males with or without DHP treatment, the GSI and sperm number analysis seems largely unaffected (Figure 2A, C, E, F, H and J). This may be the presence of androgen signaling in *ar*+/+ males, which has been reported to be necessary for proper sex differentiation and development in vertebrates.

Over long stretches, I have asked myself why the authors did generate a triple mutant rather and only a double mutant cyp17a1-/-; ngpr -/-? This double mutant appears out of the blue in figure 6; why has no other data been provided on the double mutant? It appears vital to present data on that double mutant. As stated above several times it would have been important to get an idea about the phenotype of the npgr-/- to see what happens to spermatogenesis.

Thanks for the suggestion. Please see response to essential revision 1.

Reviewer #3:Previous studies from this group showed that cyp17a1-/- fish are all male with enhanced testicular development and spermatogenesis. Since the testosterone (T) and 11-KT levels are lower in the mutant fish, it was postulated that there is an androgen-independent signaling pathway. In this study, the authors showed the levels of progestins (P4 and DHP) were elevated in cyp17a1-/- fish and treatment of ar-/- fish with exogenous P4 and DHP partially restored spermatogenesis and testicular development. By generating and analysis of cyp17a1, ar, npr double and triple mutant fish lines, they concluded that the elevated levels of progestins functions as an additional and Ar-independent pathway regulating zebrafish spermatogenesis and testicular development.Overall, the findings are new and interesting. The genetic rescue crossing data are quite convincing. The manuscript is well written and the data nicely presented.The following are my comments and questions:1) The authors showed that P4 and DHP treatment increased GSI and spermatozoa number. Did they look at the mature sperm number, their motility and fertility in the treated fish? If yes, please include the data. If not, please discuss possible outcome.

Good suggestion! Please see response to essential revision 4.

2) Is it possible to inhibit progestin synthesis/secretion in cyp17a1-/- fish or use progestin antagonists (e.g. mifepristone, ORG3170 etc.? If possible, it will provide further evidence that the elevated levels of progestins is critical.

This is a very good suggestion! Please see response to essential revision 5.

3) Figure 5 can be better described. For instance, how the authors identify SG, PSP, SSP, and SZ cells? This will help the general eLife readers. It will also be helpful to quantify the results.

According to this suggestion, the identification standards for SG, PSP, SSP and SZ cells have been provided: It has been reported that, with the differentiation of germ cells from spermatogonia to advanced stages of spermatids and sperm, Vasa signal intensity enriched in the cytoplasm decreases (Dai et al., 2021; Yu et al., 2018) (Line 201-203).

Compared with control males (Figure 5A-C), the Vasa immunofluorescence localization showed a reduced number of spermatozoa in the lumen of seminiferous tubules of *ar*-/- males (Figure 5D-F), but an increased number of spermatozoa could be observed in the lumen of seminiferous tubules of *cyp17a1*-/- fish (Figure 5G-I) and *cyp17a1*-/-;*ar*-/- fish (Figure 5J-L), again supporting that the reduced number of spermatozoa in the lumen of seminiferous tubules of *ar*-/- males was rescued by the further *cyp17a1* knockout (in the *cyp17a1*-/-;*ar*-/- fish). However, the restoration failed in the *cyp17a1*-/-;*ar*-/- fish with further depletion of *npgr* (in the *cyp17a1*-/-;*ar*-/-;*npgr*-/- fish) (Figure 5M-O), as evidenced by the greater spermatogonia, primary spermatocytes and secondary spermatocytes, but fewer spermatozoa filling in the lumen of seminiferous tubules of *cyp17a1*-/-;*ar*-/-;*npgr*-/- fish (Figure 5O) and the *ar*-/- males (Figure 5F) (Line 199-213).

Compared with the *cyp17a1*-/- fish and *cyp17a1*-/-;*ar*-/- fish, the observations of greater spermatogonia and spermatocytes, but fewer spermatozoa in the *ar*-/- males, *ar*-/-;*npgr*-/- males, and *cyp17a1*-/-;*ar*-/-;*npgr*-/- fish were observed in the histological analyses of testes (Figure 4J-Q) (Line 194-197).

4) The description and discussion about Figure 6 are not as clear as the other figures. For instance, the sycp3 results are different from those of vasa,, dnd, nano and piwill. What are these data telling us? Likewise, the patterns of star, hsd3b and cyp11a2 expression are different? What are these results saying and how do they related to the spermatogenesis phenotypes?

According to this suggestion, the transcriptome analyses of the testes of the control males, *ar*-/- males, *cyp17a1*-/- fish, *npgr*-/- males, *cyp17a1*-/-;*ar*-/- fish, *cyp17a1*-/-;*npgr*-/- fish, *ar*-/-;*npgr*-/- males, and *cyp17a1*-/-;*ar*-/-;*npgr*-/- fish, were performed and discussed (Line 215-232, 333-362). Please see response to essential revision 1.

5) Line 154: "…confirm that the accumulated progestins, P4 and DHP, acts directly to facilitate testicular development and spermatogenesis …". Please remove the word "directly". This is too strong since no data suggesting the exogenously add P4 and DHP act indirectly or indirectly.

Good suggestion! To avoid confusing the readers, we have removed the word "directly" or "direct" according to the suggestion.

[Editors' note: further revisions were suggested prior to acceptance, as described below.]

There, however, remaining issues that need to be addressed, as outlined below:1) The current version of the manuscript is very difficult to read. Many sentences are hard to understand or can be interpreted in different ways. I suggest the authors to seek help from colleagues/professionals to edit the manuscript thoroughly. They can also choose to work with an eLife copyeditor to address this issue.

We feel sorry for the language problem. As we are not native English speakers, the manuscript has been proofread by three colleagues in USA and has been double-checked by us when the manuscript was in revision. The re-worded title, the revised and detailed descriptions and discussions in the manuscript, and modifications on the figures are also made in the revised manuscript.

2) Another reason is presentational. Instead of presenting the data based on a chronological order, I suggest them to present the data following the order of Figure 7. For instance, it would be much easier for the readers to follow if they change Figure 3 into Figure 1, F1->2, and F2->.

We appreciate the suggestion. The orders of the figures have been updated according to the suggestion.

3) The fertility data should be included.

The fertility data has been included in New Figure 3—figure supplement 1. The original data has been provided in Figure 3—figure supplement 1 – Source data.

4) The RNAseq data presentation. It is clear this is a huge amount of information. What is unclear is what do these data mean? I suggest the authors to dig into this dataset and present the data in relationship to the main conclusion/model (see Figure 7). For example, is it possible to see any changes in genes involved in the progestin/androgen biosynthesis (in addition to cyp17a1), nPgr target genes, and/or Ar target genes in these mutants?While it is okay to presenting all 8 groups together, it may be more meaningful to also compare the transcriptomic changes among different pairs of genotypes (in relationship to the main conclusion in Figure 7).

Good suggestion! Accordingly, we explored the potential impact on the expression of genes in fish of different genotypes (Line 242-267). From our previous observations, the changes in genes involved in the progestin/androgen biosynthesis (gonadal steroidogenesis), including *star*, *hsd3β1*, *cyp11a2*, and *cyp11c1*, were all increased in the testes of *ar*-/- males, *npgr*-/- males, *cyp17a1*-/-;*ar*-/- fish, *cyp17a1*-/-;*npgr*-/- fish, *ar*-/-;*npgr*-/- males, and *cyp17a1*-/-;*ar*-/-;*npgr*-/- fish (Line 225-228, Figure 6A). We also analyzed the expression of genes in fish of different genotypes based on more specific comparisons. Since the normal spermatogenesis has been observed in the control males, *cyp17a1*-/-;*ar*-/- fish and *cyp17a1*-/- fish, while the defective spermatogenesis has been observed in *cyp17a1*-/-;*ar*-/-;*npgr*-/- fish. Therefore, the expressed genes in *cyp17a1*-/-;*ar*-/-;*npgr*-/- fish compared with that in the control males, *cyp17a1*-/-;*ar*-/- fish and *cyp17a1*-/- fish respectively, were analyzed and presented in a Venn diagram (Figure 6-supplement figure 1A), which could be used to identify the common transcripts responsible for the normal spermatogenesis course. Out of 1380 annotated genes, we identified a total of 148 differentially expressed genes in the overlapped region, such as the downregulated *gonadal somatic cell derived factor* (*gsdf*), *npgr*, *axonemal dynein assembly factor 3* (*dnaaf3*), *insl3* and upregulated *inhibin subunit β B* (*inhbb*) (Supplemental Table 1). The downregulated *npgr* may be resulted from the *npgr* deletion-mediated premature mRNA decay in the *cyp17a1*-/-;*ar*-/-;*npgr*-/- fish (El-Brolosy et al., 2019), and the aberrant expression of *gsdf*, *insl3* and *inhbb* may contribute to the compromised testis organization and spermatogenesis in the *cyp17a1*-/-;*ar*-/-;*npgr*-/- fish.

On the other hand, the defective spermatogenesis has been observed in the *ar*-/- males, *ar*-/-;*npgr*-/- males and *cyp17a1*-/-;*ar*-/-;*npgr*-/- fish. Therefore, the expressed genes in *ar*-/- males, *ar*-/-;*npgr*-/- males and *cyp17a1*-/-;*ar*-/-;*npgr*-/- fish compared with that in the control males respectively, were analyzed and summarized in Figure 6-supplement figure 1B, which can help to determine the common transcriptional changes responsible for the defective spermatogenesis. Out of 1315 annotated genes, we identified a total of 111 differentially expressed genes in the overlapped region, such as the downregulated *axonemal central pair apparatus protein* (*hydin*), *RNA binding motif protein 47* (*rbm47*), *axonemal dynein assembly factor 1* (*dnaaf1*), *outer dense fiber of sperm tails 3B* (*odf3b*) and *insl3* (Supplemental Table 2). These results demonstrated that phenotypes in the *ar*-/- males, *ar*-/-;*npgr*-/- males and *cyp17a1*-/-;*ar*-/-;*npgr*-/- fish may be caused by the dysregulated expressions of genes involved in the process of spermatogenesis and structure organization of sperm.

Considering the downregulation of *insl3* expression shown in the Venn diagram (Figure 6—figure supplement 1A and B), and reported in both *cyp11c1* and *cyp11a2* mutant male zebrafish, which are associated with disorganized testicular structure and significantly decreased numbers of mature spermatozoa (Li et al., 2020; Zhang et al., 2020), it is reasonable to speculate that *insl3* may be a target that is co-regulated by androgen signaling and high level of progestin signaling. Among the upstream signals of the *insl3*, the role of the accumulated progestin may be a compensatory signaling pathway that regulates testis organization and spermatogenesis in the absence of androgen signaling (Line 375-382).

During the transcriptomic analyses, we did not identify common features in the progestin/androgen biosynthesis among our various mutant genotypes. We know that the direct evidence supports the hypothesis is still lacking; therefore, the description in the Figure legends has also been corrected (Line 812-813, "The dissected regulatory network" has been changed to "The potential regulatory network"). To further our understanding of the roles of androgen and progestin in regulating testis organization and spermatogenesis via Insl3, it would be useful for future studies to obtain more genetic and biochemical evidence.